# Suppression of flavivirus transmission from animal hosts to mosquitoes with a mosquito-delivered vaccine

Dan Wen[1,2,6], Limin S. Ding [1,2,6], Yanan Zhang[1,2], Xiaoye Li[3], Xing Zhang[4], Fei Yuan[1,2], Tongbiao Zhao [5] & Aihua Zheng [1,2]✉

Zoonotic viruses circulate in the natural reservoir and sporadically spill over into human populations, resulting in endemics or pandemics. We previously found that the Chaoyang virus (CYV), an insect-specific flavivirus (ISF), is replication-defective in vertebrate cells. Here, we develop a proof-of-concept mosquito-delivered vaccine to control the Zika virus (ZIKV) within inaccessible wildlife hosts using CYV as the vector. The vaccine is constructed by replacing the pre-membrane and envelope (prME) proteins of CYV with those of ZIKV, assigned as CYV-ZIKV. CYV-ZIKV replicates efficiently in *Aedes* mosquitoes and disseminates to the saliva, with no venereal or transovarial transmission observed. To reduce the risk of CYV-ZIKV leaking into the environment, mosquitoes are X-ray irradiated to ensure 100% infertility, which does not affect the titer of CYV-ZIKV in the saliva. Immunization of mice via CYV-ZIKV-carrying mosquito bites elicites robust and persistent ZIKV-specific immune responses and confers complete protection against ZIKV challenge. Correspondingly, the immunized mice could no longer transmit the challenged ZIKV to naïve mosquitoes. Therefore, immunization with an ISF-vectored vaccine via mosquito bites is feasible to induce herd immunity in wildlife hosts of ZIKV. Our study provides a future avenue for developing a mosquito-delivered vaccine to eliminate zoonotic viruses in the sylvatic cycle.

Most zoonotic viruses, such as the Ebola virus and ZIKV, originated from wild vertebrates and have not been eradicated[1]. Zoonotic viruses are maintained mainly in vertebrates or between vertebrates and arthropod vectors[2]. Vector-borne diseases account for more than 20% of emerging/re-emerging infectious diseases[1]. Arboviruses are arthropod-borne viruses transmitted by blood-sucking arthropod vectors, such as mosquitoes and ticks[3].

ZIKV (Family *Flaviviridae*, Genus *Flavivirus*) is a positive-strand RNA virus mainly transmitted by *Aedes aegypti* (*A. aegypti*)[4]. ZIKV was first discovered in the serum of a rhesus monkey in 1947 in Uganda, Africa, while the first report from Asia was from *A. aegypti* mosquitoes in 1966 in Malaysia[5,6]. However, widespread presence of ZIKV in many other tropical African and Asian countries was revealed by subsequent serological surveys[6]. In 2007, ZIKV was reported on Yap Island[7], followed by French Polynesia in 2013[8], and Brazil in 2015, resulting in approximately 440,000 to 1,300,000 cases[9].

Similar to other flaviviruses, the ZIKV genome encodes seven nonstructural and three structural proteins[10]. The envelope (E) and

[1]State Key Laboratory of Integrated Management of Pest Insects and Rodents, Institute of Zoology, Chinese Academy of Sciences, 100101 Beijing, China. [2]CAS Center for Excellence in Biotic Interactions, University of Chinese Academy of Sciences, 100101 Beijing, China. [3]College of life sciences, Henan Normal University, 45300 Xinxiang, China. [4]College of life sciences, University of Chinese Academy of Sciences, 100049 Beijing, China. [5]State Key Laboratory of Stem Cell and Reproductive Biology, Institute of Zoology, Institute for Stem Cell and Regeneration, Chinese Academy of Sciences, Beijing, China. [6]These authors contributed equally: Dan Wen, Limin S. Ding. ✉e-mail: zhengaihua@ioz.ac.cn

membrane proteins (M) decorating the lipid membrane are the main targets of neutralizing antibodies and the important antigens for vaccine development[11]. Although most ZIKV patients have no symptoms or only mild clinical symptoms, infection during pregnancy can lead to miscarriage or Zika congenital syndrome[12]. Currently, there are no commercial vaccines or anti-viral therapies available[13].

ZIKV establishes a sylvatic transmission cycle between nonhuman primates (NHPs) and sylvatic mosquitoes in tropical Africa[14,15]. Monkeys, such as *Chlorocebus aethiops*, *Erythrocebus patas*, and *Papio papio*, serve as enzootic amplifying hosts[16,17]. However, the sylvatic ZIKV cycle in America and Asia is undefined[14,18,19].

Although genetically close to mosquito-borne flaviviruses, insect-specific flaviviruses (ISFs), such as CYV and Donggang virus (DONV), are exclusively isolated from mosquitoes and only grow in insect cells and insects[20]. Our previous study demonstrated that CYV and DONV could enter vertebrate cells but failed to initiate replication[21]. Many ISFs have been found to suppress the replication of pathogenic flaviviruses in mosquitoes[22]. ISFs are also a promising platform for flavivirus vaccine development. Chimeric Binjari virus or Aripo virus expressing the envelope proteins of ZIKV, dengue virus (DENV), or yellow fever virus (YFV) could not replicate in vertebrates, but trigger protective immune responses in vertebrates[23–25]. Using a similar strategy, a chimeric insect-specific alphavirus, Eilat virus, expressing the envelope proteins of chikungunya virus (CHIKV), elicited robust protective immunity in monkeys[26].

Here, we used CYV as a vaccine vector and constructed a chimeric vaccine by replacing the prME proteins of CYV with those of ZIKV (CYV-ZIKV). CYV-ZIKV can replicate efficiently in mosquitoes and be secreted in saliva. A protective immune response was triggered in mice after being bitten by mosquitoes carrying CYV-ZIKV, which prevented ZIKV transmission from mice to mosquitoes. Thus, we propose that releasing these vaccine-carrying mosquitoes to immunize inaccessible wild amplifying hosts is a potential approach to block the sylvatic cycle of arboviruses.

## Results

### Characterization of the CYV-ZIKV chimeric virus

Our previous study discovered that ISFs, such as CYV can enter vertebrate cells as efficiently as mosquito-borne flaviviruses but fail to initiate replication. Untranslated regions (UTRs) play a major role in the replication barrier in vertebrates[21]. In this study, we first synthesized the CYV genome in three segments and assembled them into the pACYC177 vector between a cytomegalovirus (CMV) promoter and the HDV ribozyme (RBZ) terminal site to obtain the infectious clone of CYV. To construct a CYV vectored ZIKV vaccine, the prME protein-encoding sequence of CYV was replaced with that of the MR766 strain of the ZIKV, and the resulting virus was designated as CYV-ZIKV (Fig. 1a).

CYV and CYV-ZIKV were rescued by transfecting the infectious clone plasmids into mosquito cell line C6/36. ZIKV and CYV E protein expression was detected in the C6/36 cells by immunofluorescence and in the supernatants by Western blotting using the cross-reactive monoclonal antibody (mAb) 4G2 (Fig. 1b, c). Similar growth curves of CYV-ZIKV and CYV were observed in C6/36 cells, with peak titers of $6 \times 10^8$ focus-forming units per milliliter (FFU/ml) for CYV-ZIKV and $1 \times 10^9$ FFU/ml for CYV, respectively (Fig. 1d). CYV-ZIKV grew efficiently in mosquito cells but was replication-defective in vertebrate cells, such as 293T, BHK-21, and Vero cells, as detected by immunofluorescence (Fig. 1e). The incompetence of CYV-ZIKV to produce progeny viruses ensures its safety in vertebrates.

### CYV-ZIKV could be disseminated to the saliva of *Aedes* mosquitoes after oral infection

CYV was initially isolated from *Aedes vexans* (*A. vexans*) in China. The susceptibility of the lab-adapted mosquito *A. aegypti* to CYV and CYV-ZIKV was determined by feeding the mosquitoes with blood containing $10^8$ FFU/ml viruses. High levels of CYV RNA were detected in the midgut as early as day 3 after blood feeding, indicating robust viral infection. The CYV RNA levels gradually increased over time in the fat body, leg, thorax, head, and saliva, suggesting efficient viral dissemination (Fig. 2a–f). CYV-ZIKV displayed a similar pattern of infection as the parent virus CYV, with lower RNA levels in the midgut, fat body, leg, thorax, and head but higher in the saliva (Fig. 2a–f). The mean CYV-ZIKV titer in single mosquito saliva was about 70, $6.5 \times 10^3$, and $5.0 \times 10^4$ tissue culture infectious dose ($TCID_{50}$; 50% tissue culture infectious dose) at 7-, 12-, and 19-days post-infection (dpi), respectively (Fig. 2f), with a transmission rate of 75% (Fig. S1a, b). Infection by CYV-ZIKV did not change the survival curve of *A. aegypti* compared to mock-infected mosquitoes (Fig. S1c). Another important virus vector, *Aedes albopictus* (*A. albopictus*), was also susceptible to CYV-ZIKV, with relatively lower viral RNAs in the saliva and lower transmission rates than *A. aegypti* (Fig. S2). Chimeric CYV expressing the prME of DENV4 was also successfully rescued with peak titer of $10^8$ FFU/ml in C6/36 and $10^5$ $TCID_{50}$/ml in the saliva of *A. aegypti* mosquitoes (Fig. S6). Thus, we speculate that CYV could be a universal vaccine platform for many pathogenic flaviviruses.

To rule out the possibility of CYV-ZIKV spreading to wild mosquito populations, we next investigated venereal and transovarial transmission routes. As shown in Fig. 2g, neither CYV nor CYV-ZIKV could be transmitted to naïve male mosquitoes by co-culturing with infected females. As an ISF, CYV could be efficiently transmitted via the transovarial route from infected females to F1 mosquitoes. However, the transovarial transmission of CYV-ZIKV was almost abolished (Fig. 2h), implying that the structural proteins might be responsible for the vertical transmission of CYV. Therefore, the chance for CYV-ZIKV to spread to native mosquitoes is very low. The genetic stability of CYV-ZIKV during continuous passages in mosquitoes was also evaluated. For each passage, CYV-ZIKV underwent amplification in cell culture, mosquito infection, and saliva collection. The viral titer of the infected saliva was quite stable, and only four synonymous mutations emerged in the fifth passage (Fig. 2i, j). Furthermore, the susceptibility of passage 5 (P5) of CYV-ZIKV remained the same as the parental virus in vertebrate cells at 28 °C or 37 °C (Fig. 2k). These results indicate that CYV-ZIKV did not gain the ability to infect vertebrates after serial passages.

Since *A. aegypti* mosquitoes are vectors of many pathogenic arboviruses, such as ZIKV and DENV[27], we tested the vector competence of CYV-ZIKV-carrying mosquitoes. Similar to many other ISFs[22], pre-infection with CYV or CYV-ZIKV significantly reduced ZIKV RNA levels in mosquitoes by 0.73 log or 2.23 log, respectively (Fig. S3). These results suggest that the chances for CYV-ZIKV-carrying mosquitoes serving as vectors for ZIKV are quite low.

### X-ray irradiation did not affect the CYV-ZIKV infection of *A. aegypti*

To further minimize the potential safety hazard of releasing a chimeric virus into the environment, we sterilized female *A. aegypti* mosquitoes by X-ray irradiation before viral infection. Three to four days after eclosion, female adults were collected and irradiated by X-rays at doses of 10, 20, or 40 Gray (Gy). The survival rate was above 80% for the highest dose 3 days post-exposure (Fig. 3a). After blood feeding with CYV-ZIKV, the engorged rate of all mosquitoes was above 75% (Fig. 3b). The number of eggs laid by mosquitoes and the egg hatching rate decreased significantly as the X-ray dose increased. Mosquitoes in the 40 Gy group barely laid any eggs (Fig. 3c, d). Notably, CYV-ZIKV titers in the saliva were not affected by X-ray irradiation, even at a dose of 40 Gy (Fig. 3e). These results revealed that X-rays could sterilize mosquitoes without compromising the infectivity of CYV-ZIKV in mosquitoes.

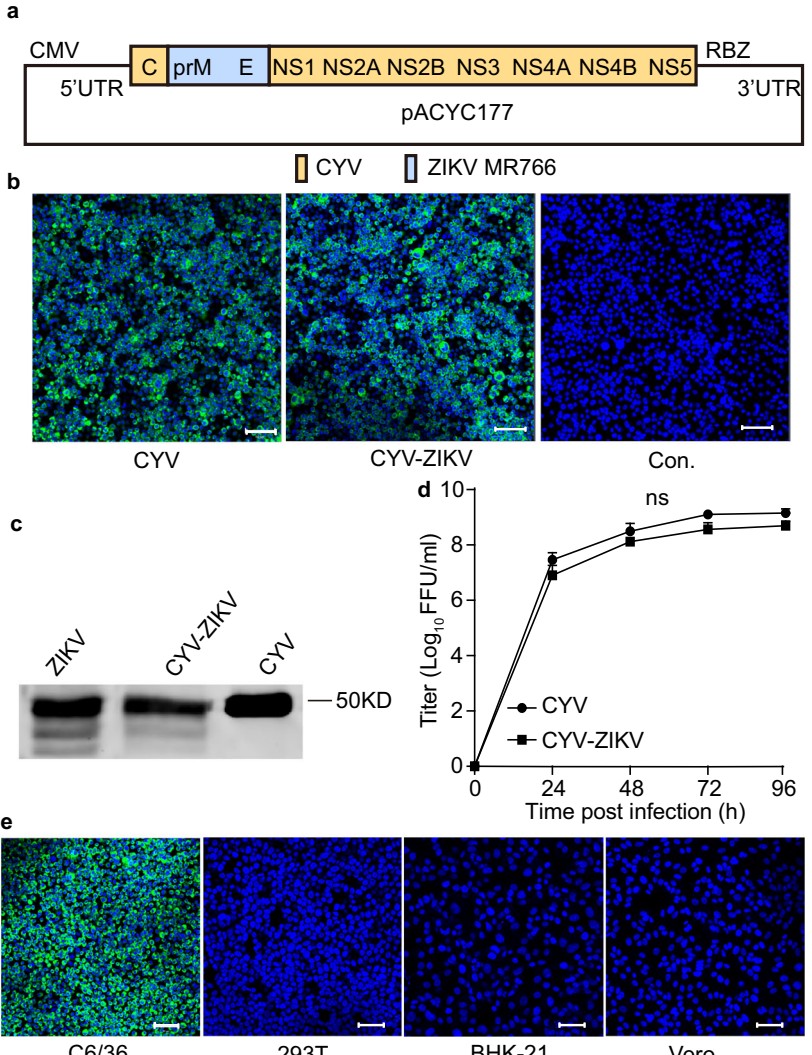

**Fig. 1 | Characterization of the CYV-ZIKV chimeric virus. a** Schematic diagram of the CYV-ZIKV chimeric virus infectious clone. The encoding sequence of the prME of the CYV (yellow) was replaced by that of the ZIKV MR766 strain (blue). The viral genome was placed between a CMV promoter and a RBZ in the pACYC177 vector. **b** The wild-type CYV and CYV-ZIKV chimeric viruses were rescued by transfection of the viral infectious clone plasmids into C6/36 cells. ZIKV E protein expression in C6/36 cells was detected by immunofluorescence with mAb 4G2 (green), and the nucleus was stained with Hoechst 33342 (blue) 3 days post-transfection. Controls (Con.) was not transfected. The bars indicate 50 μm. **c** C6/36 cells were infected with CYV, ZIKV, or CYV-ZIKV at a MOI of 1. The virus supernatants were purified by sucrose gradient centrifugation at 3 dpi, separated by western blot, and stained by anti-E mAb 4G2. The source data was provided in Fig. S10. **d** Growth curves of CYV

and CYV-ZIKV after infection of C6/36 cells at a MOI of 1. The supernatant titers were gauged with a focus-forming assay on C6/36 cells ($n = 3$). Data are presented as means ± standard deviation (SD) of triplicate measurements. The *P*-value was determined by a two-sided multiple *t*-test, and ns indicates not significant. **e** Susceptibility of CYV-ZIKV on mosquito and vertebrate cells. The cells were infected with CYV-ZIKV at a MOI of 0.1. ZIKV E protein expression in C6/36 cells was detected by immunofluorescence with mAb 4G2 (green), and the nucleus was stained with Hoechst 33342 (blue) at 3 dpi. C6/36 (*A. albopictus* cell line), 293T (human embryonic kidney cell line), BHK-21 (Baby hamster kidney cell line), and Vero (African green monkey kidney cell line). The bars indicate 50 μm. Similar results were obtained in three independent experiments. Source data are provided as a Source Data file.

## CYV-ZIKV elicited robust humoral immune responses in mice via the intraperitoneal route and mosquito bites

To test the immunogenicity of CYV-ZIKV, 6–8-week-old C57BL/6 mice were injected intraperitoneally (i.p.) with $2 \times 10^3$ FFU, $2 \times 10^5$ FFU, or $2 \times 10^7$ FFU of CYV-ZIKV and boosted with the same dosage 14 days later. Serum samples were collected at 4, 12, and 16 weeks after the first immunization to evaluate the neutralization titers against the ZIKV African lineage strain MR766 by a 50% focus reduction neutralization titer (FRNT50) assay (Fig. 4a). No weight loss was observed 1 week after the first immunization (Fig. 4b). FRNT50 titers at the fourth week were 277, 761, and 1,125 for the $2 \times 10^3$ FFU, $2 \times 10^5$ FFU, and $2 \times 10^7$ FFU dose groups, respectively. The serum neutralizing activities sustained for 16 weeks, with the FRNT50 titers decreased to different extents (Fig. 4c).

Next, we tested whether CYV-ZIKV elicits immune responses via mosquito bites. CYV-ZIKV was loaded into X-ray-irradiated *A. aegypti* by oral feeding, as described above, and $10^6$ TCID$_{50}$/ml of CYV-ZIKV was detected in the saliva 12 dpi (Fig. 4e). Mice deficient in type I IFN receptor (IFNAR$^{-/-}$ C57BL/6) were bitten by CYV-ZIKV-infected mosquitoes 1, 2, or 3 times (30 mosquitoes per mouse) at an interval of 2 weeks (Fig. 4d). Consistent with Fig. 2k, no CYV-ZIKV replication was detected at bitten sites in the mouse skin (Fig. S9). No significant weight change was observed in any of the groups after being bitten (Fig. 4f–h). The neutralizing antibodies were evaluated at week 5 after mosquito bites and all mice bitten two or three times showed seroconversion of neutralizing antibodies against the MR766. The mean neutralizing antibody titers in mice bitten 1, 2, and 3 times were 35, 987, and 158 against MR766, respectively, and 20, 534, and 78 against the

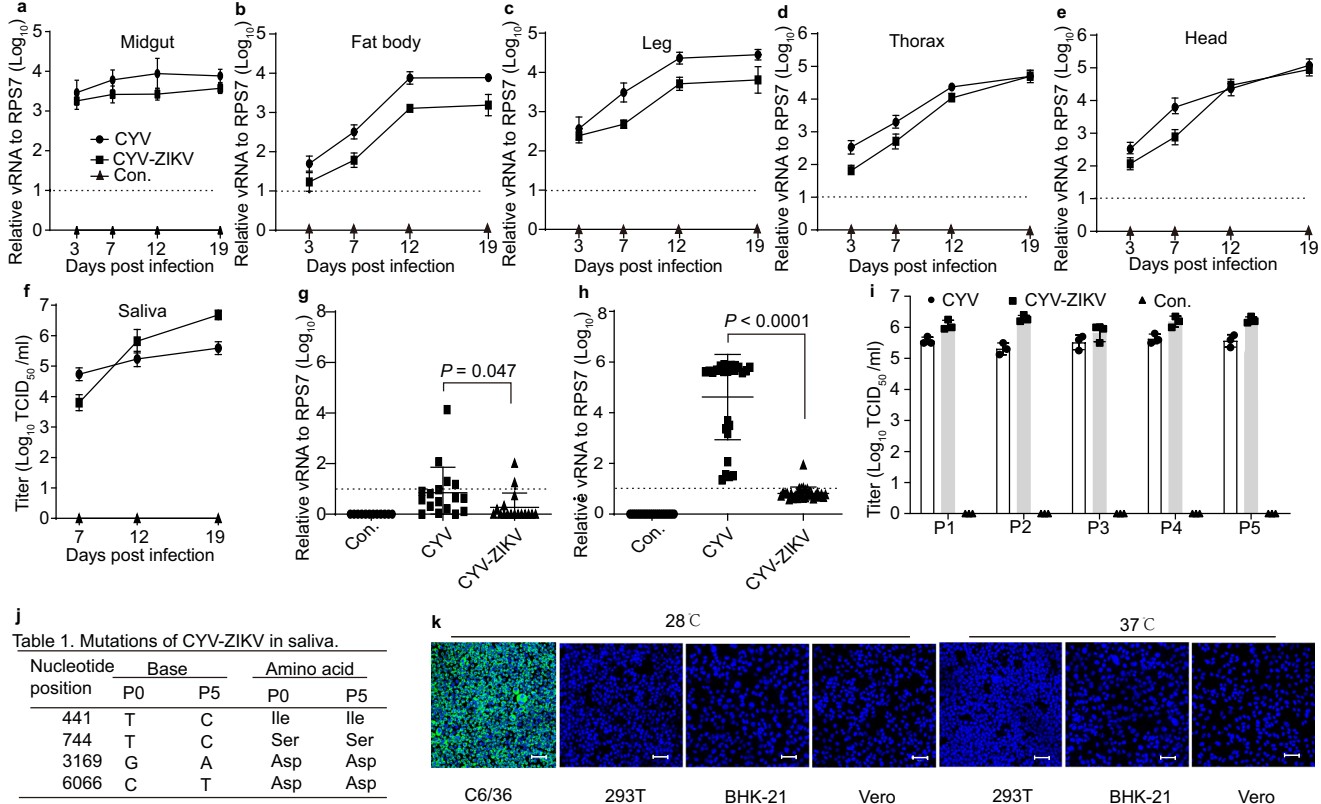

**Fig. 2 | CYV-ZIKV propagation in *A. aegypti*. a–f** Five- to six-day-old females were blood-fed with CYV-ZIKV or CYV diluted to $1 \times 10^8$ FFU/ml. Control (Con.) was fed with medium. Data are presented as means ± SD of triplicate measurements. Total RNA from the group of 20 midguts (**a**, *n* = 3), 20 fat bodies (**b**, *n* = 3), 20 legs (**c**, *n* = 3), 20 thoraxes (**d**, *n* = 3), or 20 heads (**e**, *n* = 3) were extracted at different times post-infection, and the viral RNA was detected by real-time PCR. **f** The viral titers of saliva in 80 mosquitoes were determined by $TCID_{50}$ assay (*n* = 3). **g, h** Total RNA from a single mosquito was extracted, and the viral RNA level was measured by real-time PCR. A dot represents a mosquito. Con. indicates control uninfected mosquitoes. Data are as means ± SD. The *P*-value was determined by a two-sided unpaired *t*-test. **g** Venereal transmission of the virus between females and males (*n* = 17). **h** The transovarial transmission between infected females and F1

generation (*n* = 27). The dashed line represents the detection limit. **i, j, k** Genetic stability of CYV-ZIKV in mosquitoes during passage 1 (P1) to passage 5 (P5). P0 (passage 0), P2 (passage 2), P3 (passage 3), P4 (passage 4). **i** The viral titers in the saliva of 80 mosquitoes from P1 to P5 were determined by the $TCID_{50}$ assay (*n* = 3). Data are presented as means ± SD of triplicate measurements. Con. indicates uninfected mosquito group. **j** Mutations of CYV-ZIKV emerged during passage in mosquitoes. (k) Susceptibility of CYV-ZIKV from passage 5 in mosquito (C6/36) and vertebrate cells (293T, BHK-21, and Vero cells) incubated at 28 °C or 37 °C (MOI = 0.1). The ZIKV E protein was immunolabeled with the 4G2 antibody (green staining), and the nucleus was stained with Hoechst 33342 (blue staining) at 3 dpi. The bar indicates 50 µm. Similar results were obtained in three independent experiments. Source data are provided as a Source Data file.

Asian lineage strain Natal-RGN, respectively (Fig. 4i). The serum neutralizing activities after two- or three-bite doses were markedly stronger than those after one dose, revealing that mosquito-delivered CYV-ZIKV required booster doses to elicit potent neutralizing immune responses.

Antibody-dependent enhancement (ADE) caused by antibodies targeting prM on partially matured viral particles is an important safety concern for vaccine development against ZIKV and its antigenically related DENV[28,29]. To determine the maturation extent of CYV-ZIKV in mosquito saliva, we evaluated prM-specific antibody levels in CYV-ZIKV-immunized mice via mosquito bites compared to i.p. injection. Mouse sera samples with similar neutralizing activity from the two groups (Fig. 4c, j) were selected to perform enzyme-linked immunosorbent assay using the recombinant ZIKV prM protein (Fig. S8a, b). Comparable levels of anti-prM antibodies were detected (Fig. S8c), suggesting that the maturity of CYV-ZIKV was consistent regardless of being amplified in mosquitoes or in C6/36 cells. We further tested whether immunization of CYV-ZIKV caused ADE against DENV using K562 cells expressing an Fc-receptor and no ADE effect was detected (Fig. S7).

We further investigated the effect of the number of mosquitoes per dose on the immune response. IFNAR$^{-/-}$ C57BL/6 mice were bitten by 3, 10, or 20 CYV-ZIKV-carrying mosquitoes 3 times at an interval of

2 weeks. Neutralizing antibodies in the sera against the MR766 strain were determined 5 weeks after the first bite. The mean FRNT50 titers were 127 for the 3 mosquitoes group, 588 for the 10 mosquito group, and 136 for the 20 mosquito group. However, ZIKV-specific neutralizing antibodies were only detected in 40% of the mice bitten by 3 mosquitoes (Fig. 4j). Notably, being bitten three times by 10 mosquitoes per mouse induced neutralizing antibodies cross-reactive against African lineage MR766, Asian lineage Natal-RGN, and GZ01 strains, which persisted for up to 5 months (Fig. S4).

## CYV-ZIKV-mosquito bites protected IFNAR$^{-/-}$ C57BL/6 mice from the ZIKV challenge

IFNAR$^{-/-}$ C57BL/6 mice are an established infection model for ZIKV and other flaviviruses[30]. We first tested the 50% lethal dose (LD$_{50}$) of three ZIKV strains in IFNAR$^{-/-}$ C57BL/6. As shown in Fig. S5, the LD$_{50}$ was less than 20 FFU for MR766 and GZ01 and $1 \times 10^4$ FFU for Natal-RGN. Viral titers in the saliva of 80 mosquitoes used to immunize mice were above $10^6$ $TCID_{50}$/ml at 12 dpi (Fig. 5b–d). Based on the above study, we chose three doses (10 mosquitoes per animal) with an interval of 2 weeks as the immunization protocol. At 5 weeks after the first immunization, the mean FRNT50 titers were 704 against MR766 (Fig. 5e), 94 against Natal-RGN (Fig. 5i), and 225 against GZ01 (Fig. 5m). To further evaluate the vaccine efficacy, immunized mice were i.p.

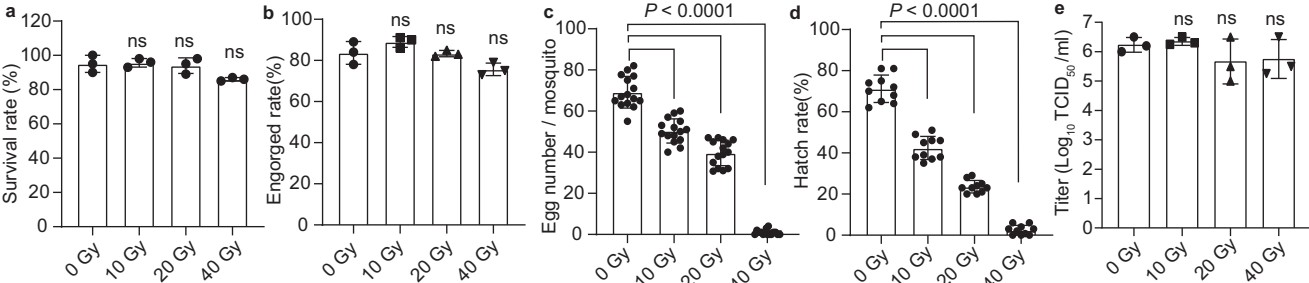

**Fig. 3 | Propagation of CYV-ZIKV in X-ray irradiated *A. aegypti* mosquitoes.** Two- to three-day-old female adults were X-ray-irradiated at various doses, and then blood-fed with CYV-ZIKV diluted to $1 \times 10^8$ FFU/ml at 3–4 d post-irradiation. **a** Survival rate of 30 adults at 7 d post-irradiation ($n = 3$). Error bars are presented as means ± SD of triplicate measurements. **b** Engorged rate of 30 adults ($n = 3$). Data are presented as means ± SD of triplicate measurements. **c** The number of eggs laid per mosquito at 4 days post blood-feeding ($n = 15$).

Data are presented as means ± SD. **d** The hatching rate of the eggs ($n = 10$). Data are presented as means ± SD. **e** Viral titers in the saliva of 80 mosquitoes at 12 dpi ($n = 3$). Error bars are presented as means ± SD of triplicate measurements. The *P*-value was determined by a two-sided unpaired *t*-test and compared with the group of 0 Gy. ns indicates not significant. Similar results were obtained in three independent experiments. Source data are provided as a Source Data file.

challenged with $10^3$ FFU (>50 LD$_{50}$) of the MR766 strain, $10^5$ FFU (10 LD$_{50}$) of the Natal-RGN strain, or $10^4$ PFU of GZ01 (>5 × $10^2$ LD$_{50}$) strain at 42 days post-first bitten. Survival rate (Fig. 5f, j, n), weight loss (Fig. 5g, k, o), and ZIKV viremia (Fig. 5h, l, p) were monitored for 14 days. Notably, 100% of vaccinated animals survived the lethal virus challenge with no or less than 10% weight loss and a significant decrease in viremia. However, all control mice bitten with naïve mosquitoes died after challenge, except for one animal in the Natal-RGN group (Fig. 5j). Hence, CYV-ZIKV delivered by mosquito bites could render complete and cross-reactive protection in the mouse model.

## Mosquito vaccines blocked the transmission of ZIKV from host to vector

Since ZIKV is circulated between wild vertebrate hosts and sylvatic mosquitoes in a natural epidemic focus[2,31], we further investigated ZIKV transmission from vaccinated animal hosts to mosquitoes. CYV-ZIKV was first loaded into *A. aegypti* by oral infection, as described above, and $10^6$ TCID$_{50}$/ml of CYV-ZIKV was detected in the saliva at 12 dpi (Fig. 6b). Groups of IFNAR$^{-/-}$ C57BL/6 mice were bitten by these mosquitoes three times (10 mosquitoes per animal) every 2 weeks and then challenged with $10^3$ FFU of the ZIKV MR766 strain 14 days later. The mean neutralization antibody titer of the CYV-ZIKV-immunized mice sera was 437 at 5 weeks after the first bite (Fig. 6c). Mouse weight was monitored for 3 days post-challenge and slight weight loss was observed in the group bitten by the CYV-ZIKV mosquitoes compared with the obvious weight loss in the control group (Fig. 6d). ZIKV viremia in the CYV-ZIKV group was ~4 log lower than that in the control group, as measured by focus-forming assay at day 3 post-challenge (Fig. 6e). On the same day, 5–6-day-old *A. aegypti* mosquitoes were blood fed on the mice. The viral RNA load of individual mosquitoes was assessed by real-time PCR at 7 days post-feeding. As expected, the mean ZIKV RNA levels in mosquitoes fed with the blood of CYV-ZIKV-immunized mice were lower than the detection limit, while most mosquitoes fed with the control mice (Con.) were positive for ZIKV (Fig. 6f). These results suggest that mosquito-delivered CYV-ZIKV vaccines can successfully block ZIKV transmission from hosts to naïve mosquitoes.

## Discussion

Herd immunity achieved by vaccination has been proven effective in controlling infectious diseases, such as smallpox, measles, and poliomyelitis, in the urban community[32,33]. However, zoonotic viruses are difficult to eradicate due to their natural foci and sylvatic circulation. For example, CHIKV is circulated between NHPs and arboreal *Aedes* spp. mosquitoes in the sylvatic cycle in Africa[34]. In Senegal, continuous surveillance of CHIKV seroprevalence in NHPs and virus prevalence in

mosquitoes revealed that CHIKV epidemics in the enzootic African cycle are periodic and negatively related to NHP herd immunity[35,36]. Population immunity based on a high seroprevalence also plays an important role in limiting ZIKV spread in urban areas[37,38]. Thus, it is rational to prevent zoonotic spillovers by vaccinating animal reservoirs.

Wildlife immunization has been applied in only a few zoonotic diseases and most are delivered by oral baits, among which recombinant live rabies vaccines is well established[39]. Field trials of oral attenuated classical swine fever vaccine have also been performed in Europe since the 1990s at large scale[40]. The barriers limiting wildlife vaccination include: (i) involvement of multiple hosts in sylvatic transmission cycles; (ii) safety concerns for non-target species; (iii) high reproductive rates and population turnover; (iv) fastidious behaviors and difficulty in designing effective delivery systems; (v) difficult delivery due to extreme low or high population densities of the target hosts; (vi) environmental concerns for the release of genetically modified organisms; (vii) stability of a vaccine under prevailing environmental conditions; and (viii) low unit cost for vaccine purchase and delivery[41].

In the sylvatic cycle, the amplifying hosts of ZIKV, DENV, YFV, and CHIKV are mainly monkeys[2,16,42]. Monkeys are social animals with low density, mostly distributed in specific nature reserves, and the range of their activities is relatively concentrated[43]. According to the International Union for Conservation of Nature (IUCN), about 75% of primates have a declining population, with 60% of primates facing extinction. The behavior and distribution of NHPs are relatively well investigated due to their close genetic relationship with humans[44,45]. NHPs are long-lived enzootic hosts with a low reproducing rate (for example, 0.42 per female per year for howler monkeys)[46]. Therefore, it would be plausible to achieve and maintain herd immunity for a long period in NHPs. Viruses with small natural foci would be eliminated by herd immunity. For widespread viruses, belt-immunization around a human community could be applied to prevent spillover from sylvatic circulation.

Release of sterile mosquitoes and *Wolbachia*-carrying mosquitoes have been studied for many years and shown significant efficacy in controlling mosquito populations in field trial[47–49]. Sustained releases of transgenic *A. aegypti* males with the OX513A lethal gene led to at least 80% suppression of the wild *A. aegypti* population in the Cayman Islands in 2010 and a suburb of Juazeiro, Bahia, Brazil in 2012[50–52]. Releasing *Wolbachia*-infected male mosquitoes successfully reduced wild *A. albopictus* populations on two separate islands in Guangzhou, China, from 2014 to 2017 and wild *A. aegypti* populations in Australia from 2017 to 2018[48,49]. Thus, mass production and release of mosquitoes is feasible for the control of arboviruses.

Achieving herd immunity in vertebrates through mosquito bites would be a considerable approach. In our study, we developed this

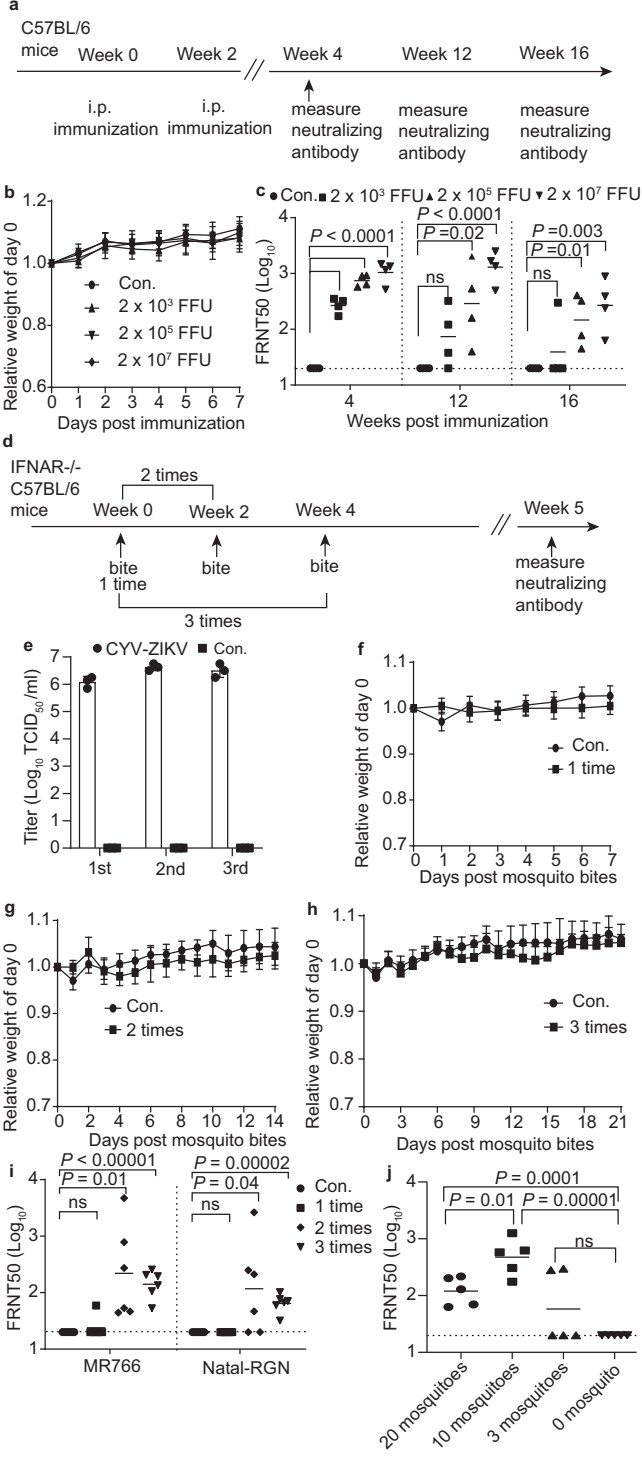

**Fig. 4 | Immunogenicity of CYV-ZIKV in mice by direct injection and mosquito bite. a**–**c** C57BL/6 mice were immunized with two doses of the indicated doses of CYV-ZIKV or PBS (Con.) via the i.p. route (*n* = 4). **a** Scheme of the immunization experiment. **b** Body weight was monitored for one week. Data are indicated as means ± SD. **c** Neutralizing antibody level of sera was tested on 4-, 12-, and 16-week post-vaccination by FRNT50 assay against the ZIKV MR766 strain. The dashed lines indicate the 1:20 detection limit and the horizontal bars the mean. The *P*-value was determined by a two-sided multiple *t*-test. **d**–**i** IFNAR⁻/⁻ C57BL/6 mice (CYV-ZIKV, *n* = 6; Con., *n* = 5) were bitten with CYV-ZIKV-infected or naïve *A. aegypti* mosquitoes (30 per mouse) one to three times. **d** The process of mosquito bites and bleeding of mice. **e** Viral titer in the saliva of 80 mosquitoes used to immunize mice at 12 dpi (*n* = 3). 1st, the first bite; 2nd, the second bite; 3rd, the third bite. Con. indicates uninfected mosquito group. Data are presented as means ± SD of triplicate measurements. **f**–**h** Body weight was monitored after mosquito bites. Con. indicates control mice (*n* = 5) bitten by naïve mosquitoes. 1 time, *n* = 6; 2 times, *n* = 6; 3 times, *n* = 6. Data are defined as means ± SD. **i** Neutralizing antibody levels of sera were evaluated by a FRNT50 assay against ZIKV MR766 or Natal-RGN strains at 5 weeks post-first bite. Con. indicates control mice bitten by naïve mosquitoes. The dashed lines indicate the 1:20 detection limit, and the horizontal bars indicate the mean. The *P*-value was determined by a two-sided multiple *t*-test. **j** IFNAR⁻/⁻ C57BL/6 mice were bitten by 0, 3, 10, or 20 mosquitoes three times with an interval of 2 weeks (*n* = 5). ZIKV MR766-specific neutralizing antibody titers were tested by FRNT50 assay. The dashed lines indicate the 1:20 detection limit, and the horizontal bars indicate the mean. The *P*-value was determined by a two-sided multiple *t*-test. Similar results were obtained in three independent experiments. Source data are provided as a Source Data file.

ISFs are effective vaccine vectors for flaviviruses, with the potential of a human clinical trial. Vaccines against ZIKV, DENV, WNV, and JEV are developed using ISF vectors, such as Binjari and Aripo viruses[23,25]. Here, CYV-ZIKV also elicited robust humoral immune responses in IFNAR⁻/⁻ mice via mosquito bites and lasted for at least 5 months, with no ADE against DENV observed in vitro, which needs to be further confirmed in vivo. (Fig. S7). Mice bitten three times were fully protected against ZIKV challenges by various strains with significantly decreased viremia, which blocked the transmission of ZIKV to naïve mosquitoes. However, a higher humoral immunity response was observed in mice receiving two bites rather than three bites (Fig. 4i), while the immunogenicity of CYV-ZIKV was dose-dependent as administered by the i.p. route (Fig. 4c). We assume that components in the saliva might suppress the immune responses, which could be overcome by increasing the titer of CYV-ZIKV in the saliva.

The ISF-vectored CYV-ZIKV vaccine is safe regarding target hosts and environment. CYV-ZIKV replicates efficiently in mosquitoes but is replication-deficient in vertebrates with a known mechanism[21]. No side effects were observed after immunization into mice, either through mosquitoes or needles. Similar results were achieved by ZIKV vaccines developed by other ISF vectors, such as the Binjari virus, immunized via the intramuscular route[23]. Furthermore, CYV-ZIKV is contained in released mosquitoes and is unable to spill over into the environment venereally to male mosquitoes or transovarially spread to the new generation of mosquitoes. The mosquitoes were further sterilized by X-ray without affecting the CYV-ZIKV titers in the saliva. The genome of CYV-ZIKV was very stable without amino acid mutation after 5 serial passages in mosquito saliva. CYV-ZIKV suppressed the ZIKV infection efficiency in mosquitoes released by more than 2 log. Thus, released vaccine-carrying mosquitoes have very few biosafety risks in nature.

Arboviruses kill millions of wild animals[55]. Targeted immunization of wild animal reservoirs is a promising approach, not only for control of zoonotic diseases infecting domestic animals and humans, but also for protection of endangered wildlife, such as Ruffed grouse decreased by West Nile virus. Our study provides a future avenue for developing a mosquito-delivered vaccine for wildlife immunization.

"mosquito vaccine" with lab-adapted *A. aegypti*. Although *A. aegypti* mosquitoes were considered a human vector, a recent field study revealed that *A. aegypti* mosquitoes also feed on wild and domestic animals in South Florida, USA[53]. Furthermore, lab-adapted *A. aegypti* feeds on a wild range of animals, varying from birds to mammals, including NHPs[54]. CYV vector shows high prevalence in *Aedes* mosquitoes and CYV-ZIKV was highly susceptible in *A. aegypti*, resulting in high and sustained titer in the saliva. Thus, we believe that the lab-adapted *A. aegypti* could be a promising vector for our vaccine due to its high susceptibility and simple maintenance. In our study, CYV-ZIKV also infected the temperate and tropical mosquito *A. albopictus*, which further expanded the application range of CYV-ZIKV.

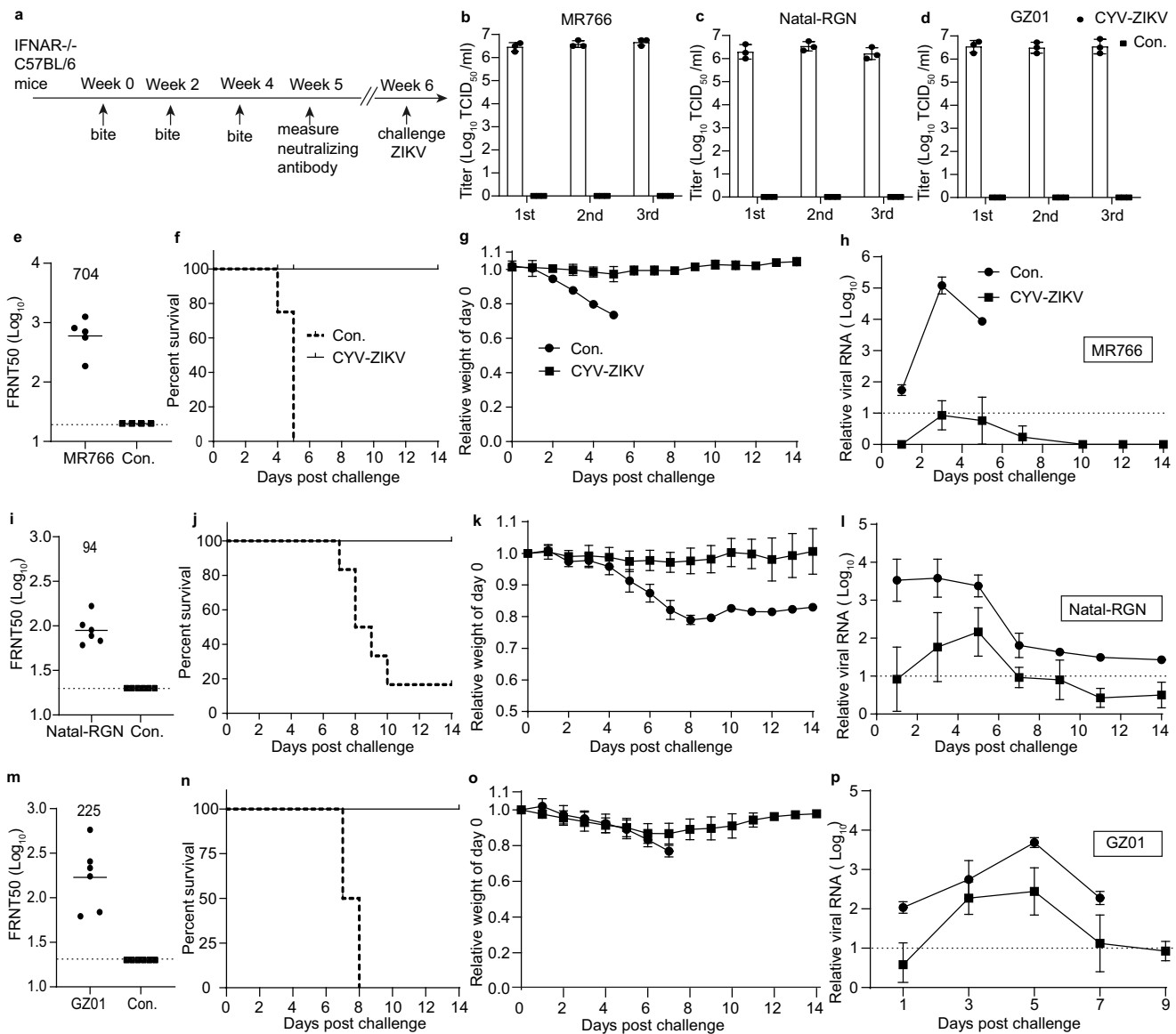

**Fig. 5 | CYV-ZIKV-mosquitoes bites protect IFNAR$^{-/-}$ C57BL/6 mice from the ZIKV challenge. a–p** Mice were bitten with CYV-ZIKV infected mosquitoes (10 per mouse) three times and challenged on day 42 with $10^3$ FFU (>50 LD$_{50}$) of the ZIKV MR766 strain (**f–h**), $10^5$ FFU (10 LD$_{50}$) of the Natal-RGN strain (**j–l**), or $10^4$ PFU (>5 × $10^2$ LD$_{50}$) of the GZ01 strain (n-p) ($n = 6$). Mice bitten with naïve mosquitoes were used as controls (Con.). **a** Scheme of the immunization and challenge experiment. **b–d** Viral titers of CYV-ZIKV in the saliva of 80 mosquitoes used to immunize mice against ZIKV MR766 (**b**), Natal-RGN (**c**), or GZ01(**d**) were determined 12 dpi ($n = 3$). 1st, the first bite; 2nd, the second bite; 3rd, the third bite. Data are presented as means ± SD of triplicate measurements. Con. indicates control saliva of naïve mosquitoes. **e, i, m** Neutralizing antibody levels of sera on day 35 against the indicated ZIKV strain. The dashed lines indicate the 1:20 detection limit and the horizontal bars indicate the mean. **e** MR766, $n = 5$; Con., $n = 4$. **i** $n = 6$. **m** $n = 6$. **h, j, n** Survival rate of mice. **g, k, o** Weight change in mice. Data are presented as means ± SD. **h, l, p** ZIKV viremia post-challenge, measured by real-time PCR using ACTB as the reference gene. The dashed lines indicate the detection limit. Data are presented as means ± SD. Similar results were obtained in three independent experiments. Source data are provided as a Source Data file.

## Methods

### Ethics statement

All mice and mosquito experiments were performed strictly following bioethics principles and were supervised by the Bioethics Committee of the Institute of Zoology, Chinese Academy of Science (IOZ-IACUC-2020-067). ZIKV experiments were performed under biosafety level 2 (BSL2) and animal BSL2 (A-BSL2) containment. Mice were housed under the following conditions: ambient temperature 22 ± 1°C, humidity control 50%, 12 h light/12 h dark cycle.

### Cells and antibodies

Vero cells (ATCC, CCL-81), BHK-21 cells (ATCC, CCL-10) and 293 T cells (ATCC, CRL-3216) were maintained in DMEM plus 8% fetal bovine serum (FBS) and 1% L-glutamine at 37 °C with 5% CO$_2$. C6/36 cells

(ATCC, CRL-1660) were maintained in RPMI 1640 plus 8% heat-inactivated FBS and 1% L-glutamine at 28 °C with 5% CO$_2$. K562 cells (ATCC, CCL-243) were maintained in RPMI 1640 plus 8% heat-inactivated FBS and 1% L-glutamine at 37 °C with 5% CO$_2$. 4G2 is a mAb that recognizes the fusion peptide of the E protein of all flavi-viruses, including DENV and ZIKV[56,57].

### Construction of CYV, CYV-ZIKV, and CYV-DENV4 infectious clones

The CYV genome was synthesized using NC_017086 (GenBank sequence accession number; https://www.ncbi.nlm.nih.gov/nuccore/NC_017086.1/) as the template. The complete genome was divided into three overlapping segments, including A (positions 1 to 2472), B (positions 2473 to 6468), and C (positions 6469 to 10733), and

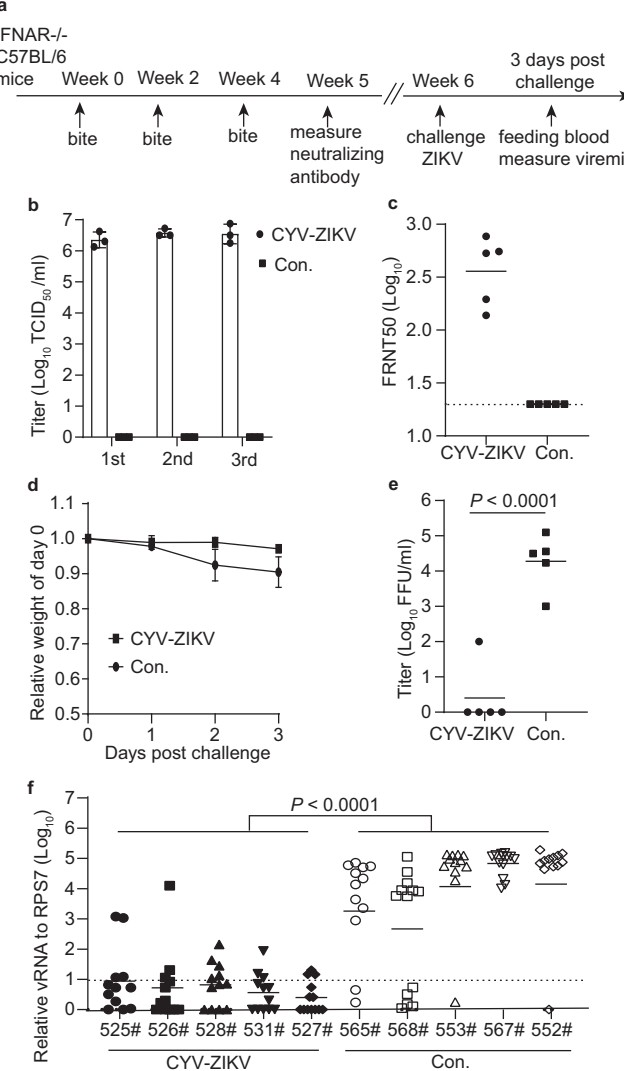

**Fig. 6 | Mosquito vaccines can block the transmission of ZIKV from mice to mosquitoes. a** Scheme of the vaccination and challenge and oral infection schedule. **b** Viral titers of CYV-ZIKV in the saliva of 80 mosquitoes were determined 12 dpi before the immunization of mice ($n = 3$). Con. indicates control saliva of naïve mosquitoes. Data are presented as means ± SD of triplicate measurements. **c** IFNAR$^{-/-}$ C57BL/6 mice were bitten by 10 CYV-ZIKV-infected mosquitoes or naïve mosquitoes (Con.) per animal three times with an interval of 2 weeks ($n = 5$). ZIKV MR766-specific neutralizing antibody titers were tested by a FRNT50 assay on day 35. The dashed line indicates the detection limit of 1:20 and the horizontal bars indicate the mean. **d** Mice ($n = 5$) were i.p. challenged with $10^3$ FFU of the ZIKV MR766 strain. Weight change was monitored for 3 days. Data are as means ± SD. **e** Viremia at 3 days post-challenge was measured using the focus-forming assay ($n = 5$). Data are as means ± SD. The $P$-value was determined by a two-sided unpaired $t$-test. **f** Naïve mosquitoes were fed on mice ($n = 5$) for 20 min at 3 days post-challenge. Total RNA from a single mosquito was extracted at 7 days post-feeding (525#, 526#, 528#, 531#, $n = 12$; 527#, 565#, 568#, 553#, 567#, 552#, $n = 13$), and the viral RNA was detected by real-time PCR and then normalized against the reference gene RPS7. Con. indicates control mosquitoes fed on naïve mice. The x axis represents the serial number of mice. The dashed lines indicate the detection limit. Data are presented as means ± SD. The $P$-value was determined by the two-sided nested $t$-test. Similar results were obtained in three independent experiments. Source data are provided as a Source Data file.

synthesized by Beijing SYKM Gene Biotechnology. Three segments were further assembled into low-copy-number vector pACYC177 with a CMV promoter in front of the 5′ terminus and a hepatitis delta virus (HDV) ribozyme (RBZ) terminal site following the 3′ end

using an assembly SoSoo mix (Cat. TSV-S2, Beijing TsingKe Biotech, China).

To generate CYV-ZIKV or CYV-DENV4, the CYV prME genes were replaced with the prME genes from ZIKV MR766 strain (GenBank sequence accession number, HQ234498; https://www.ncbi.nlm.nih.gov/nuccore/HQ234498) or DENV4 1228 strain (GenBank sequence accession number, KX239897; https://www.ncbi.nlm.nih.gov/nuccore/KX239897). The prME genes of ZIKV MR766 (amino acid positions 123 to 799) were amplified using the Zika MR766 infectious clone[58]. The prME genes of DENV4 were synthesized by Beijing SYKM Gene Biotechnology. The prME segments and CYV infectious clone segments were assembled using SoSoo mix (Cat. TSV-S2, Beijing TsingKe Biotech, China).

*Escherichia coli* strain Stbl3 (Cat. BC108-01, Biomed, China) was used for all the transformations and cultured at 30 °C. These infectious cDNA clones were fully sequenced to ensure the absence of any mutations.

### Rescue of infectious clones
Recombinant CYV viruses were rescued by transfection of the infectious clone plasmid into C6/36 cells with FuGENE® 6 transfection reagent (Cat. E2691, Promega, USA). After 5 to 6 days, the supernatant of C6/36 cells was collected and stored at −80 °C.

### Immunofluorescence assay
Virus-infected cells were fixed with 4% paraformaldehyde for 30 min at room temperature. PBS plus 0.5% Triton X-100 was used for permeabilization at room temperature for 10 min. The cells were blocked with 3% BSA in PBS at room temperature for 1 h. Subsequently, the cells were incubated with mouse mAb 4G2 (Cat. E2691, GeneTex, China) at a dilution of 1:120 overnight at 4 °C to detect the flavivirus E protein. After being washed with PBS 3 times, cells were incubated with Alexa Fluor 488 goat anti-mouse secondary antibody diluted at 1:400 (Cat. A11001, Invitrogen™, USA) for 45 min in the dark at room temperature. Hoechst 33342 (1 μg/ml) (Cat. C0031, Solarbio Life Sciences, China) was added at room temperature at 1 g/ml to stain the nuclei. Fluorescence was collected from Zeiss Zen 2010 software (version 6.0; Zeiss, Germany) by confocal microscopy (Zeiss LSM 710, Germany).

### Mosquito infection experiment
*A. aegypti* (UGAL/Rockefeller strain) and *A. albopictus* (Jiangsu strain) mosquitoes were maintained in the laboratory. Larval mosquitoes were fed with mice food, albumin, and yeast extract. Adult mosquitoes were provided with water and 10% (wt/vol) sucrose. Five- to six-day-old females (150 for each group) were starved for 20 h before oral infection. Before feeding, the viruses from C6/36 supernatant were diluted with 300 μl RPMI, mixed with 300 μl mouse blood, and preheated at 37 °C for 30 min. Then, mosquitoes were fed on the mixtures through a very thin parafilm with a circulating water system for 30 min at 37 °C. After feeding on the blood, the mosquitoes were chilled at 4 °C for 10 min, and the engorged mosquitoes were collected and maintained on a sugar solution.

### Focus-forming assay
The C6/36 or vero cells were seeded in 96-well plates at 40,000 or 8000 cells per well at 24 h before the experiment. Virus supernatants were 10-fold serial diluted in RPMI medium plus 2% FBS and infected cells for 2 h (100 μl/well). Then, the mediums with virus were replaced with fresh RPMI plus 2% FBS, 1% penicillin/streptomycin, and 20 mM $NH_4Cl$. After incubation at 28 °C for 3 days, the cells were fixed, permeabilized, and blocked as described above in the immunofluorescence assay. Virus foci were stained with anti-E antibody (4G2) followed by Alexa Fluor 488 goat anti-mouse secondary antibody and counted under a fluorescence microscope.

## Collection of saliva from single mosquito

Seventeen-day-old female mosquitoes (12 for each group) were anesthetized in a 4 °C refrigerator, then legs and wings were removed on ice. The mouthpart was inserted into a 20-μl pipette tip containing 5 μl of FBS for 30 min to collect the saliva. Saliva samples were diluted with 50 μl of RPMI medium containing 2% FBS, centrifuged at 13,000 x g for 30 min, and then filtered with a 0.22-μm filter. The virus titer in single saliva was detected by focus-forming assay.

## Tissue culture infectious dose immunofluorescence assay for virus titer in saliva

Female mosquitoes (80 for each group) were fed on the 800 μl RPMI medium containing 2% FBS at 37 °C for 30 min with a circulating water system to collect saliva from a group of mosquitoes. The saliva was centrifuged at 13,000 x g for 30 min and filtered with a 0.22-μm filter.

The virus titers of saliva were determined by $TCID_{50}$ immunofluorescence assay. In brief, the C6/36 cells were passaged in 96-well plates at 40,000 cells per well. At cell confluency 80–90% the next day, 10-fold serial diluted virus supernatants were added to the plate (100 μl/well) with eight repeats. After incubation at 28 °C for 3 days, the infection was detected by immunofluorescence using the 4G2 antibody. The titers were expressed as $\log_{10} TCID_{50}$/ml and calculated using the Reed–Muench method[59].

## RNA extraction and real-time PCR analysis

RNA from mosquito heads, thorax, midguts, fat bodies, and legs (20 for each group) was extracted using Trizol reagent (Cat. 15596018, Invitrogen™, USA). RNA from the blood and virus supernatant was extracted using the TIANamp Virus RNA Kit (Cat. DP315-R, TIANGEN, China). Quantification of viral and housekeeping gene RNA was performed using a One-Step SYBR PrimerScript reverse transcription (RT)-PCR protocol (Cat. RR066A, Takara Bio Inc, Japan). CYV and CYV-ZIKV RNA were detected using the following primers: sense, 5′GCT GCTGTGAAAGGCAACAAGTCTG3′; antisense, 5′GACTCCAGCACTCC TCTTCCCC3′. ZIKV MR766 RNA was quantified using the following primers: sense, 5′GGGGAAACGGTTGTGGACTT3′; antisense, 5′CTGGG AGCCATGCACTGATA3′. ZIKV GZ01 RNA was quantified using the following primers: sense, 5′GACATGGCTTCGGACAGCCG3′; antisense, 5′ CTTAGCGCATGTCACCAGGCTC3′. ZIKV Natal-RGN RNA was quantified using the following primers: sense, 5′ CACTTGAAATGTCGCCTGA A3′; and antisense, 5′TCCCTGCGTACTGTACCTCC3′. The sequences of the RPS7 (mosquito housekeeping gene ribosomal protein gene S7) primers were as follows: sense, 5′TCAGTGTACAAGAAGCTGACCGG A3′; antisense, 5′TTCCGCGCGCGCTCACTTATTAGATT3′. The ZIKV viral RNA levels from mice blood were normalized against reference gene β-actin (ACTB). The sequences of the ACTB primers were as follows: sense, 5′ATCGTGCGTGACATCAAAGAG3′; antisense, 5′ATGCCAC AGGATTCCATACCC 3′. All of these real-time PCR data were collected and analyzed from QuantStudio 12k Flex software (version 1.4; ABI, USA) by ABI QuantStudio 12k Flex Real-Time PCR System (Applied Biosystems, Life Technologies, Carlsbad, CA, USA).

## Serial passage of CYV-ZIKV in saliva

After feeding on the mixture of CYV-ZIKV and mice blood for 12 days, the saliva of female mosquitoes (80 for each group) was added to C6/36 cells for viral amplification as the first generation. Then, mosquitoes were fed on the first-generation virus and blood mixture through a very thin Parafilm from a circulating water system for 30 min at 37 °C. The saliva viruses were amplificated as above until the fifth-generation virus was successfully amplified.

## Immunization of C57 mice by i.p. injection

Specific-pathogen-free (SPF) C57BL/6 WT mice were purchased from Beijing Vital River Laboratory Animal Technology (licensed by Charles River). Six- to eight-week-old male and female C57BL/6 mice (4 for each group) were immunized with $2 \times 10^3$ FFU, $2 \times 10^5$ FFU, $2 \times 10^7$ FFU CYV-ZIKV or PBS through the i.p. injection route. Two weeks later, the mice were boosted through the same route and immunization dose. After immunization, a weight change was detected in the mice. At 1, 3, and 4 months after i.p. injection, mice were bled to measure neutralizing antibody titers using a FRNT50 assay.

## Immunization of IFNAR$^{-/-}$ C57BL/6 mice by mosquito bites

Female *A. aegypti* mosquitoes were X-ray irradiated at a dose of 40 Gy and orally infected by CYV-ZIKV from C6/36 supernatant. IFNAR$^{-/-}$ C57BL/6 mice were purchased from the Institute of Laboratory Animal Science, Chinese Academy of Medical Science & Peking Union Medical College. On weeks 0, 2, and 4, mosquitoes (15 dpi) bite 6–8-week-old male and female IFNAR$^{-/-}$ C57BL/6 mice (5–6 for each group) for 20 min. Before immunization, mosquito saliva was collected to detect the viral load in saliva at 12 dpi. In week 5, mouse sera were separated to detect neutralizing antibodies by a FRNT50 assay.

## Neutralization assay

The neutralizing activity of mouse sera was assessed using ZIKV MR766, Natal-RGN (GenBank sequence accession number, KU527068; https://www.ncbi.nlm.nih.gov/nuccore/KU527068), and GZ01 strain (GenBank sequence accession number, KU820898; https://www.ncbi. nlm.nih.gov/nuccore/KU820898). The vero cells were seeded in 96-well plates at 8,000 cells per well at 24 h before the experiment. Sera samples were three-fold serially diluted starting at 1:20 in DMEM with 2% FBS and 1% penicillin/streptomycin. The diluted sera were incubated with the same volume of 100 FFU ZIKV at 37 °C for 30 min. The antibody and ZIKV mixtures were added to the Vero cells in 96-well plates for 2 h. Then, the mixtures were removed and replaced with DMEM plus 2% FBS, 1% penicillin/streptomycin, and 20 mM $NH_4Cl$. Vero cells were incubated at 37 °C for 3 days. The cells were then fixed, permeabilized, and blocked as described in immunofluorescence assay. Virus foci were stained with anti-E antibody (4G2) followed by Alexa Fluor 488 goat anti-mouse secondary antibody, and the fluorescence was observed under a fluorescence microscope. The results were quantified as the $FRNT_{50}$.

## Mice infection with ZIKV

6–8-week-old IFNAR$^{-/-}$ C57BL/6 male and female mice (4 for each group) were challenged with $10^3$ FFU of the ZIKV MR766 strain (>50 $LD_{50}$), $10^5$ FFU of the ZIKV Natal-RGN strain (10 $LD_{50}$), or $10^4$ PFU of the ZIKV GZ01 strain (>5 × $10^2$ $LD_{50}$) by i.p. at two sites in a volume of 100 μl each site at 42 days post-immunization and monitored for weight loss, survival rate, and viremia.

## Mosquito irradiation

Eighty *A. aegypti* female adults (2–3 days old) were collected and irradiated at 10 Gy, 20 Gy, and 40 Gy using an RS 2000 series Biological Irradiator (Rad Source Technologies Inc. USA.). After irradiation, the mosquitoes were transferred to a cage and supplied with 10% w/v sucrose solution at 28 °C. These mosquitoes were starved for 20 h, and the survival rates were recorded before oral infection 3–4 days after irradiation. The engorged rate, number of mosquitoes laying eggs, egg hatching rate, and viral titer of saliva were recorded after oral infection.

## Venereal transmission experiment

Seventeen three-day-old female *A. aegypti* mosquitoes were intrathoracically infected with 60 FFU viruses and mated with the same number of healthy males at 3 dpi. Mated male mosquitoes were sacrificed for RNA extraction and real-time PCR of a single male to detect the RNA level of the virus at 10 dpi.

## Transovarial transmission experiment

Fifty three-day-old female *A. aegypti* mosquitoes were intrathoracically infected with 60 FFU viruses and mated with the same number of healthy males at 3 dpi. Mated female mosquitoes were blooded with pure mice blood without virus after 7 days. Fully engorged females (30–40) were individually transferred into a moistened cage with 5% sucrose cotton wool and bred for oviposition. Then, females that laid eggs were tested for viral infection by real-time PCR, defining as $F_0$. Offsprings in eggs stage of virus positive $F_0$ were collected and fed for eclosion. Single mosquitoes (27 for each group) were sacrificed respectively to detect the transovarial infection rate by real-time PCR at 5 days post eclosion.

## Protein expression and purification

The coding sequence of ZIKV MR766 prM (1–124 aa) was cloned into the pET28a vector by homologous recombination. The recombinant prM with a C-terminal His-tag was expressed in *Escherichia coli* Rosetta DE3 (Cat. MCC0050, Frdbio Bioscience & Technology, China) in the presence of 0.1-mM isopropyl-β-ᴅ-thiogalactopyranoside (IPTG) at 16 °C for 20 h. Then, the prM was purified through Ni-chelating affinity chromatography and stored at −80 °C.

## Western blot analysis of viral particles

C6/36 cells were infected with CYV, ZIKV, or CYV-ZIKV at a MOI of 1, and 10 ml of supernatants were collected at 3 dpi. Viral particles were precipitated by ultra-centrifugation through a 20% sucrose cushion by 187,000×*g* (SW41 Ti rotor, Beckman, Fullerton, CA, USA) for 3 h at 4 °C, and pellets were resuspended in 100 μl PBS. Samples were separated by 10% SDS-PAGE and immunoblotted with anti-E mAb 4G2, and the blots were scanned from Image Studio software (version 5.2; LI-COR, USA) by LI-COR Odyssey CLx Infrared Imaging System (LI-COR Biosciences, Lincoln, NE, USA).

## Enzyme-linked immunosorbent assay

96-well plates were coated with recombinant ZIKV prM protein diluted in sodium carbonate-sodium bicarbonate buffer (pH 9.6) by 100 ng/well at 4 °C overnight. Plates were blocked with 1% BSA in PBS at 37 °C for 2 h. Mouse sera were 3-fold serially diluted from 1:180 in PBS and then added into the plate (100 μl/well) at 37 °C for 2 h, following with HRP-conjugated Affinipure Goat Anti-Mouse IgG (H + L) (Cat. SA00001-1, Proteintech, USA) at 1:1000 dilution for 1 h. Soluble TMB substrate solution (Cat. PA107-01, TIANGEN, China) was added to incubate for 5 min, and sulfuric acid was then added to stop the reaction. Absorbance values at 450 nm were measured from SoftMax Pro 7 software (version 7.1; Molecular Devices, USA) by SpectraMax i3 multi-mode microplate reader (Molecular Devices, San Jose, CA, USA).

## Antibody-dependent enhancement analysis

The K562 cells were seeded in 24-well plates (10⁶ cells/well). Mouse sera were 3-fold serial diluted from 1:4 to 1:972 in DMEM and incubated with equal volume of DENV1 West Pacific strain (GenBank sequence accession number, U88535; https://www.ncbi.nlm.nih.gov/nuccore/U88535), DENV2 New Guinea C strain (GenBank sequence accession number, M29095; https://www.ncbi.nlm.nih.gov/nuccore/M29095) or DENV3 H87 strain (GenBank sequence accession number, KU050695; https://www.ncbi.nlm.nih.gov/nuccore/KU050695) (10⁵ FFU) for 1 h at 37 °C, and then transferred to K562 cells. Viral RNAs in the supernatants were measured by real-time PCR at 4 dpi.

## CYV-ZIKV replication in mice

Six- to eight-week-old male and female IFNAR⁻/⁻ C57BL/6 mice (3 for each group) were anesthetized and 1 cm² of the abdominal fur was shaved while the surrounding area was covered with paper. The shaved skin was bitten by 30 CYV-ZIKV-carrying mosquitoes per mouse for 20 min. The skin tissues at the bitten site were dissected after scarification 12, 24, 48, 72, or 96 h post bite. The viral RNA of CYV-ZIKV in the skin was determined by real-time PCR using ACTB gene as reference.

## Viral growth kinetics

The C6/36 cells were passaged in 6-cm dishes at a density of $3 \times 10^6$ cells/dish. Twenty-four hours later, CYV or CYV-ZIKV was added to the cells at a MOI of 1. The supernatants were collected at 0, 24, 48, 96 h post infection and stored at −80 °C. The C6/36 cells were seeded in 96-well plates at 40,000 cells per well. Twenty-four hours later, virus supernatants were 10-fold serially diluted with RPMI medium containing with 2% FBS and then added into C6/36 cells (100 μl/well). After incubation at 28 °C for 3 days, viral titers were determined by focus-forming assay.

## Statistics and reproducibility

All graphs and statistical analysis were generated with GraphPad Prism v.8.3.0 software (GraphPad Software, San Diego, CA). Error bars of all figures represent as mean ± SD. A two-sided unpaired *t* test or two-sided multiple *t* test was applied to calculate the significance as indicated in each figure legend. A *P* value of ≤0.05 is considered statistically significant. Sample size was determined based on statistical analysis. No data were excluded. Mice and new adult insects were randomly allocated to the experiments. No blinding occurred during these studies. Experimental design did not require blinding because assessed variables are not confounded by the evaluator. We only focused on measurable variables (weight, number of eggs, survival rate, etc).

## Reporting summary

Further information on research design is available in the Nature Portfolio Reporting Summary linked to this article.

# Data availability

Sequences information used in our work were all acquired from NCBI database, including Chaoyang virus gene (Genbank sequence accession number: NC_017086), Zika virus MR766 strain sequence (HQ234498), ZIKV Natal-RGN (KU527068), ZIKV GZ01 (KU820898), DENV1 West Pacific strain (U88535), DENV2 New Guinea C strain (M29095), DENV3 H87 strain (KU050695), and DENV4 1228 strain (KX239897). Plasmids used in the study were freely available upon request. No custom code or mathematical algorithm were used in this work. All data were available within this article, as well as supplementary information or source data files. All protocols have been described in Methods or in references therein. Source data are provided with this paper.

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

## Acknowledgements

We thank Dr. Qiyong Liu from the Chinese Center for Disease Control and Prevention for providing *A. albopictus* and Dr. Chengfeng Qin from the Academy of Military Medical Science for providing the ZIKV GZ01 strain. This work was supported by grants to A.Z. This work was funded by the National Natural Science Foundation of China Major Program (32090024), National Key R&D Program of China 2021YFC2300903, Strategic Priority Research Program of the Chinese Academy of Sciences (Grant No. XDPB16), National Natural Science Foundation of China General Program (81871687), key program of the Chinese Academy of Sciences (KJZD-SW-L11), and Open Research Fund Program of the State Key Laboratory of Integrated Pest Management (IPM1806 and IPM1603).

## Author contributions

A.Z. and F.Y. designed the study. A.Z. supervised the whole project. D.W., L.D., X.L., Y.Z., X.Z., and T.Z. performed the experiments. A.Z. and F.Y. wrote the manuscript.

## Competing interests

The authors declare no competing interests.
