## [Peer review file · Nature Communications]

REVIEWER COMMENTS

Reviewer #1 (Remarks to the Author):

This manuscript by Wen et al. describes the use of the insect-specific flavivirus, Chaoyang virus, to develop a vaccine for Zika virus that is transmitted by mosquitoes. The goal is to use these mosquitoes to vaccinate zoonotic hosts such as nonhuman primates. Insect-specific arboviruses including flaviviruses have been described several times before as vaccine candidates, so the main novelty in this paper is their transmission to vertebrates via mosquito feeding. Overall, the paper appears to demonstrate that the chimeric virus infects mosquitoes and can be delivered orally to mice to induce a protective immune response (although several of the figure legends do not include enough detail or proper statistical analysis to fully evaluate the data). The authors propose that the release of sterilized mosquitoes infected with the chimeric virus can be used to eliminate natural circulation of Zika and other arboviruses. Theoretically this might be possible, but the regulatory hurdles to releasing a genetically engineered virus into nature, as well as the logistic challenges of producing and releasing enough mosquitoes to affect enzootic circulation that is very widespread in Africa, are understated and deserve additional discussion. Also, the rationale described here is to immunize zoonotic hosts, but the experimental work is all done with human rather than enzootic vectors (however it can be considered a proof-of-principle study). The other major problem with the paper is that it does not cite previous work developing flavivirus and alphavirus insect-specific viruses as vaccine candidates.

Specific comments:

1. The paper needs editing to correct grammatical errors
2. Line 63: There is little or no evidence that *Aedes albopictus* transmits Zika virus
3. Line 64: Phylogenetic studies indicate that Zika virus spread to Asia many decades ago, before 1947
4. Line 72: Many other congenital malformations occur, thus the term “Zika congenital syndrome”
5. Line 85: Additional descriptions and references should be added for chimeric insect-specific arboviruses including flaviviruses and alphaviruses developed previously as vaccines
6. Paragraph beginning at line 89: This information on vector control does not directly relate to the work described and should be condensed or removed.
7. Line 123 and elsewhere, spell out abbreviated names such as FFU the first time they are used.
8. Line 272: In Senegal, *A. aegypti formosus* is not considered an important enzootic vector for Zika and other arboviruses, and its susceptibility to some arboviruses is very different than that of *A. aegypti aegypti*.

9. Figure 1 and others: the controls need more explanation, and in general there is not enough information in the legends to understand all of the experiments. Error bars are not defined and statistics not explained in many panels.
10. Figure 2: Panel H appears to show one infected sample. The legend is not clear but if this is one mosquito infected transovarially it is inconsistent with line 148.
11. Figure 4: A multiple test correction is needed for the statistics; also Fig. S2, S3.

Reviewer #2 (Remarks to the Author):

The manuscript by Wen and colleagues describes the production of a chimeric virus using the backbone of the insect-specific virus, Chaoyang virus and the structural prM/E proteins of Zika virus, to generate a potent vaccine candidate that protects mice against ZIKV challenge and is transmissible via the bite of mosquito.

Several reports of the construction ISF-ZIKV chimeras have been previously published along with their use as a vaccine to induce protective antibody responses in mouse models of the disease – so this is not novel or original. Indeed, the studies with the Binjari-ZIKV chimeras were far more thorough in this context and included evidence of fetal protection, high resolution structural and antigenic analysis of the chimeric virus. Such analyses were not performed for the CYV-ZIKV chimera in this study.

However, dissemination of the CYV-ZIKV chimeric virus in *Aedes aegypti* and a high rate (75%) of transmission in orally inoculated mosquitoes is new and interesting and given that CYV has surprisingly been detected in a variety of mosquito genera previously provides insight on how it has reached multiple mosquito genera. The fact that WT CYV could be transmitted transovarially is also new, but not discussed. The fact that CYV is vertically transmitted and not CYV-ZIKV – are you suggesting that it is the structural proteins of the chimera that are blocking the vertical transmission of CYV-ZIKV?. Vertical transmission of a DISF has not been previously demonstrated.

That the virus reaches sufficient titre in salivary glands and allows immunisation of mice through mosquito bites (3 mozzies x 3 times) is really surprising considering there could be little more (and probably less) than 1 ng of viral antigen present in the inoculum AND considering the virus does not show evidence of replicate in vertebrate cultures (and presumably the mouse). Nevertheless this was consistent with mice receiving a similarly low dose (10³ infectious units by IP with no adjuvant) also seroconverting with a neutralising antibody response. In this context further confirmation that there was no replication on the bitten/inoculated mice would have been useful here. Indeed, a further

dissection of the mechanisms of the potency of the response to this virus in this context would have been far more compelling and useful than the claim that his provided a basis to control arbovirus transmission in sylvatic/zoonotic cycles (see below).

Based on the immune response generated in the mosquito-bitten mice, it was no surprise that they became partially resistant to subsequent infection and failed to transmit the virus to feeding mosquitoes. This is interesting and novel data. However, the main message delivered by the authors appears to be that this is proof of concept for a “feasible” and “promising” use of this approach to immunise wildlife to interfere with sylvatic and zoonotic transmission cycles to prevent/reduce human infection, was far too speculative and optimistic in my mind and needs to be discussed with much more circumspection, if mentioned at all. There are just far too many variables for an approach like this work in a natural environment (multiple vectors of these viruses in sylvatic cycles with variable host preference, requirement for repeated release in remote regions, let alone getting regulatory approval for an uncontrolled release of GMO virus that is transmitted to animals (and humans that get in the way). There was also no repeated passage of the virus in vertebrate systems to demonstrate the virus does not adapt to vertebrate replication through selected mutations.

Specific comments:

P3, para 3 – refs and full stop required at end of this sentence

“Chimeric ISFs expressing the envelope proteins of pathogenic flaviviruses do not replicate in vertebrates but can trigger a protective immune response in vertebrates”

Several places - Binj change to Binjari;

There needs to be much more detailed methodology, which can easily be provided as a supplementary file. In particular, greater detail is required for the following to allow accurate review of the data:

The neut method (below) is rather unusual and readout and interpretation criteria very sketchy for such a crucial part of the paper. There needs to be more detail and (published) evidence that it is comparable to standard neut assays in this context. Specifically, there also needs to be information on how many units of ZIKV were added into the neutralisation assays. The addition of NH₄Cl would block the maturation of secreted virions, thus allowing for individual infected cells to be identified. I am not familiar with this type of assay.

The neutralizing activity of mouse sera was assessed using ZIKV MR766, Natal-RGN, and GZ01. Sera samples were three-fold serially diluted starting at 1:20 in DMEM with 2% FBS and 1% penicillin/streptomycin. The diluted sera were incubated with the same volume of ZIKV at 37°C for 30

min. The antibody and ZIKV mixtures were added to the Vero cells in 96-well plates for 2 h. Then, the mixtures were removed and replaced with DMEM plus 2% FBS, 1% penicillin/streptomycin, and 20 mM NH₄Cl. Vero cells were incubated at 37°C for 3 days. The neutralizing antibody was detected using the immunofluorescence assay described above.

The TCID₅₀ immunofluorescence assay needs to be defined. Given that the readout is FFU, is this assay an immunoplaquing assay? Has this method been published previously?

A Western blot is provided in Figure 1, with not methodology provided. Furthermore, there is no discussion on why the banding pattern of CYV-ZIKV differs from that of the WT ZIKV. I suspect that this could be differences in glycosylation, but it is not clear why WT differs from the chimera.

For the mouse immunization experiments, were the mice given chimeric virus that had been purified, or as a culture supernatant? Similarly for the mosquito experiments, were the mosquitoes provided purified virus, or virus as culture supernatant.

Please provide accession numbers for all of the ZIKV strains.

Method are missing for virus purification and growth kinetics.

Introduction – references are missing for other previously characterised dISF-based chimeric ZIKV vaccines – BinJV-ZIKV and ARPV-ZIKV

Reviewer #3 (Remarks to the Author):

The manuscript from Wen, Ding et al reports on the use of mosquitoes infected with an insect flavivirus chimera as a novel vaccine delivery system. The results are noteworthy as they are the first report of such a strategy and if the approach can be shown to be safe and effective could be used to reduce viral loads in animal reservoirs. The data presented is robust and convincing, but some additional work could improve the impact (discussed below). Methodology is sound and the authors should be congratulated on including the use of multiple ZIKV strains in the in vitro analysis and in vivo studies. Major and minor points listed below.

Q1 How dependent on the use of MR766 is the immunogenicity? CYV-ZIKV (MR766) had increased levels in the saliva suggesting altered tropism. This may be different for various chimeric viruses. Have the authors tested any other chimeric viruses? This would greatly improve the impact of the paper.

Q2 The authors report a lower neutralization level for the mice exposed to the most mosquitos (20 vs 10, Fig4J). Can the authors speculate on the reason for this? The memory response is also worse in the mice exposed to infected mosquitoes 3 times vs 2 times against the divergent ZIKV strains and should be mentioned and discussed.

Q3 Are the particles effectively matured? The maturation state of expectorated virus should be assessed. This could be done by western blot, or alternatively the level of prM specific antibody could be evaluated in mice bitten by the CYV-ZIKV infected mosquitos and compared to CYK-ZIKV immunized mice. Maturation state is important for ADE potential.

Q4 ADE should be evaluated. Do the mice become sensitive to dengue infection? Or does the sera from mice exposed to CYK-ZIKV cause ADE in vitro? This is very important when considering the targeted reservoirs will likely be exposed to more than one circulating flavivirus and multiple DENV serotypes.

ZIKV/Zika virus and CYV/Chaoyang are used interchangeably throughout – please use abbreviation once defined.

Line 87 missing full stop and reference needed

Line 161 are/where quite low

Line 245 suggest reword “hardly”

Line 280 define BinJ

Line 306 “A lethal gene..” sentence is out of place and lacks context. At least include a reference.

Line 324 second definition of mAb not required

Line 329 The CYV virus genome was synthesized using...

Line 437 the neutralization level was determined using immunofluorescence

Fig S1 legend is confusing, assuming the authors mean the mosquitoes where infected with 1×10^8 /ml on line 710. Was this blood fed?

Line 713, this is a mean across three independent experiments?

REVIEWER COMMENTS

Reviewer #1 (Remarks to the Author):

This manuscript by Wen et al. describes the use of the insect-specific flavivirus, Chaoyang virus, to develop a vaccine for Zika virus that is transmitted by mosquitoes. The goal is to use these mosquitoes to vaccinate zoonotic hosts such as nonhuman primates. Insect-specific arboviruses including flaviviruses have been described several times before as vaccine candidates, so the main novelty in this paper is their transmission to vertebrates via mosquito feeding. Overall, the paper appears to demonstrate that the chimeric virus infects mosquitoes and can be delivered orally to mice to induce a protective immune response (although several of the figure legends do not include enough detail or proper statistical analysis to fully evaluate the data). The authors propose that the release of sterilized mosquitoes infected with the chimeric virus can be used to eliminate natural circulation of Zika and other arboviruses. Theoretically this might be possible, but the regulatory hurdles to releasing a genetically engineered virus into nature, as well as the logistic challenges of producing and releasing enough mosquitoes to affect enzootic circulation that is very widespread in Africa, are understated and deserve additional discussion. Also, the rationale described here is to immunize zoonotic hosts, but the experimental work is all done with human rather than enzootic vectors (however it can be considered a proof-of-principle study). The other major problem with the paper is that it does not cite previous work developing flavivirus and alphavirus insect-specific viruses as vaccine candidates.

Answer:

These are excellent suggestions. We agree there are a lot of concerns about releasing vaccine-carrying mosquitoes. So, we added more discussions.

Several field releases of mosquitoes have been performed with promising results, suggesting mass production and release of mosquitoes is feasible for the control of arboviruses. We added one paragraph in the discussion as “Release of sterile mosquitoes and *Wolbachia*-carrying mosquitoes have been studied for many years and shown significant efficacy in controlling mosquito populations in field trial. Sustained releases of transgenic *A. aegypti* males with the OX513A lethal gene led to at least 80% suppression of the wild *A. aegypti* population in the Cayman Islands in 2010 and a suburb of Juazeiro, Bahia, Brazil in 2012. Releasing *Wolbachia*-infected male mosquitoes successfully reduced wild *A. albopictus* populations on two separate islands in Guangzhou, China, from 2014 to 2017 and wild *A. aegypti* populations in Australia from 2017 to 2018. Thus, mass production and release of mosquitoes is feasible for the control of arboviruses.”

We admit the mosquito-delivered vaccine is still at the early age of development. Many issues need to be addressed before releasing this kind of vaccine in a natural environment. We summarized these issues in the discussion as “The barriers limiting wildlife vaccination include: (i) involvement of multiple hosts in sylvatic transmission cycles; (ii) safety concerns for non-target species; (iii) high reproductive rates and population turnover; (iv) fastidious behaviors and difficulty in designing

effective delivery systems; (v) difficult delivery due to extreme low or high population densities of the target hosts; (vi) environmental concerns for the release of genetically modified organisms; (vii) stability of a vaccine under prevailing environmental conditions; and (viii) low unit cost for vaccine purchase and delivery.”

Other than that, the mosquito-delivered vaccine could have more usages. The mosquito-delivered vaccine could also be applied to protect endangered animals such as Ruffed grouse decrease due to West Nile virus infection (Nemeth NM, et al. West Nile virus infection in Ruffed grouse (*Bonasa umbellus*) in Pennsylvania, USA: a multi-year comparison of statewide serosurveys and vector indices. *J Wildl Dis.* 2021 Jan 6;57(1):51-59. doi: 10.7589/JWD-D-19-00016. PMID: 33635996.). Recently, a clinical trial was performed to evaluate a genetically engineered *Plasmodium falciparum* parasite vaccine delivered by mosquito bites (Murphy SC, et al. A genetically engineered *Plasmodium falciparum* parasite vaccine provides protection from controlled human malaria infection. *Sci Transl Med.* 2022 Aug 24;14(659):eabn9709. doi: 10.1126/scitranslmed.abn9709. Epub 2022 Aug 24. PMID: 36001680.)

In terms of the choice of mosquito vector, we made use of lab-adapted *A. aegypti* to deliver vaccine. Although *A. aegypti* mosquitoes are considered as human vector. However, recent study reveals that *A. aegypti* mosquitoes are also feed on wild and domestic animals in South Florida (Olson MF et al. High Rate of Non-Human Feeding by *A. aegypti* Reduces Zika Virus Transmission in South Texas. *Viruses.* 2020 Apr 17;12(4):453. doi: 10.3390/v12040453. PMID: 32316394.). Furthermore, lab-adapted *A. aegypti* feeds on a wild range of animals varying from birds to mammals including monkey. Thus, the lab-adapted *A. aegypti* could be a good vector for our vaccine due to the high susceptibility and easy maintenance.

We cited the papers about developing arbovirus vaccines with insect specific viral vectors as answered below.

Specific comments:

1. The paper needs editing to correct grammatical errors

Answer:

Thank you for pointing this out. We have corrected grammatical errors in this paper.

2. Line 63: There is little or no evidence that *Aedes albopictus* transmits Zika virus

Answer:

Thank you for pointing this out. We deleted “*Aedes albopictus*” in Page 2, Line 65.

3. Line 64: Phylogenetic studies indicate that Zika virus spread to Asia many decades ago, before 1947

Answer:

We modified as “ZIKV was first discovered in the serum of a rhesus monkey in 1947 in Uganda, Africa, while the first report from Asia was from *A. aegypti* mosquitoes in 1966 in Malaysia. The widespread presence of ZIKV in many other tropical African and Asian countries was revealed by subsequent serological surveys.” Page 2, Line 65.

4. Line 72: Many other congenital malformations occur, thus the term “Zika congenital syndrome”

Answer:

Thank you for pointing out. We corrected as suggested in Page 3, Line 77.

5. Line 85: Additional descriptions and references should be added for chimeric insect-specific arboviruses including flaviviruses and alphaviruses developed previously as vaccines.

Answer:

We summarized the insect-specific flavivirus- and alphavirus- based vaccines and cited three papers as below in Page 3, Line 90. Chimeric Binjari virus or Aripo virus expressing the envelope proteins of ZIKV, dengue virus (DENV), or yellow fever virus (YFV) could not replicate in vertebrates, but trigger protective immune responses in vertebrates (Hobson-Peters J, et al. A recombinant platform for flavivirus vaccines and diagnostics using chimeras of a new insect-specific virus. *Sci Transl Med.* 2019 Dec 11;11(522):eaax7888; Harrison JJ, et al. Chimeric Vaccines Based on Novel Insect-Specific Flaviviruses. *Vaccines (Basel).* 2021 Oct 22;9(11):1230.). Using a similar strategy, a chimeric insect-specific alphavirus, Eilat virus, expressing the envelope proteins of chikungunya virus (CHIKV), elicited robust protective immunity in monkeys (Erasmus JH, et al. A chikungunya fever vaccine utilizing an insect-specific virus platform. *Nat Med.* 2017 Feb;23(2):192-199).

6. Paragraph beginning at line 89: This information on vector control does not directly relate to the work described and should be condensed or removed.

Answer:

This is a good suggestion. We condensed this paragraph and moved it to the discussion in Page 7, Line 301.

7. Line 123 and elsewhere, spell out abbreviated names such as FFU the first time they are used.

Answer:

Thank you for pointing out. We spell out abbreviated names such as FFU (Page 3, line 119), TCID₅₀ (Page 4, line 134), FRNT50 (Page 5, line 184) the first time they are used.

8. Line 272: In Senegal, *A. aegypti formosus* is not considered an important enzootic vector for Zika and other arboviruses, and its susceptibility to some arboviruses is very different than that of *A. aegypti aegypti*.

Answer:

We agree with you that *A. aegypti formosus* is not a competent vector for many pathogenic arboviruses (Aubry F et al. Enhanced Zika virus susceptibility of globally invasive *Aedes aegypti* populations. *Science.* 2020 Nov 20;370(6519):991-996. doi: 10.1126/science.abd3663. PMID: 33214283.). So, we deleted this sentence and added one paragraph in Page 7, Line 310 “Achieving herd immunity in vertebrates through mosquito bites would be a considerable approach. In our study, we developed this “mosquito vaccine” with lab-adapted *A. aegypti*. Although *A. aegypti* mosquitoes were considered as human vector, recent field study reveals that *A. aegypti* mosquitoes also feed on wild and domestic animals in South Florida, USA (Olson MF et al. High Rate of Non-Human Feeding by *Aedes aegypti* Reduces Zika Virus Transmission in South Texas. *Viruses.* 2020 Apr 17;12(4):453. doi: 10.3390/v12040453. PMID: 32316394.). Furthermore, lab-adapted *A. aegypti*

feeds on a wild range of animals varying from birds to mammals including NHPs (Macdonald WW. Host feeding preferences. Bull World Health Organ. 1967;36(4):597-9. PMID: 4964871.).”

9. Figure 1 and others: the controls need more explanation, and in general there is not enough information in the legends to understand all of the experiments. Error bars are not defined and statistics not explained in many panels.

Answer:

Thank you for pointing out. We added more details, defined the error bars and explained statistics in the legends of figure 1-6 and supplementary figure 1-9.

We added “Controls (Con.) was not transfected.” on line 753, “Control (Con.) was fed with medium” on line 770, “Con. indicates uninfected mosquito group.” on line 788 and 819 and other explanation of controls on line 841, 854 and 864.

We added more details in the legends such as below.

“(d) Growth curves of CYV and CYV-ZIKV after infection of C6/36 cells at a MOI of 1. The supernatant titers were gauged with a focus-forming assay on C6/36 cells (n = 3). Error bars indicate the standard deviation (SD). The *P*-value was determined by a multiple *t*-test and ns indicates not significant. (e) Susceptibility of CYV-ZIKV on mosquito and vertebrate cells. The cells were infected with CYV-ZIKV at a MOI of 0.1. ZIKV E protein expression in C6/36 cells was detected by immunofluorescence with mAb 4G2 (green), and the nucleus was stained with Hoechst 33342 (blue) at 3 dpi. The bars indicate 50 μm. These results are representative of three independent experiments.” (Figure 1, page 17).

“(g) Venereal transmission of the virus between females and males. Naïve unmated males were co-cultured with virus-infected females in the same cage at 3 dpi and tested for viral RNA at 10 dpi. (h) Three-day-old female mosquitoes were intrathoracically infected with 60 FFU of viruses and mated with naive males at 3 dpi. Mated females were fed with mouse blood and fully engorged individuals were transferred into a moistened cage for oviposition. Five-day-old F1 generation was sacrificed to detect the viral RNA by real-time PCR.” (Figure 2, page 19).

“The dashed lines indicate the 1:20 detection limit, and the horizontal bars indicate the mean. The *P*-value was determined by a multiple *t*-test. **P* < 0.05. ***P* < 0.01. ****P* < 0.001. *****P* < 0.0001.” on line 823 (Figure 4).

“The *P*-value was determined by a multiple *t*-test. **P* < 0.05. ***P* < 0.01. ****P* < 0.001. *****P* < 0.0001.” on line 885 (Figure S2).

10. Figure 2: Panel H appears to show one infected sample. The legend is not clear but if this is one mosquito infected transovarially it is inconsistent with line 148.

Answer:

We have corrected as “However, the transovarial transmission of CYV-ZIKV was also almost abolished” in Page 4, Line 148.

11. Figure 4: A multiple test correction is needed for the statistics; also Fig. S2, S3.

Answer:

We corrected as suggested in Figure 4 (Page 22), Fig. S2(Page 27) and Fig. S3 (Page 28).

Reviewer #2 (Remarks to the Author):

The manuscript by Wen and colleagues describes the production of a chimeric virus using the backbone of the insect-specific virus, Chaoyang virus and the structural prM/E proteins of Zika virus, to generate a potent vaccine candidate that protects mice against ZIKV challenge and is transmissible via the bite of mosquito.

Several reports of the construction ISF-ZIKV chimeras have been previously published along with their use as a vaccine to induce protective antibody responses in mouse models of the disease – so this is not novel or original. Indeed, the studies with the Binjari-ZIKV chimeras were far more thorough in this context and included evidence of fetal protection, high resolution structural and antigenic analysis of the chimeric virus. Such analyses were not performed for the CYV-ZIKV chimera in this study.

Answer:

We agree with the reviewer that Binjari-ZIKV chimera vaccine have more thorough results on fetal protection, high resolution structural and antigenic analysis. However, here we mainly focused on the delivery of vaccine by mosquitoes and blocking of virus transmission between vertebrate hosts and mosquito vectors. Besides, we resolved high-resolution structure of insect-specific flavivirus Donggang virus, which is very close to ZIKV in our previous paper, thus we did not study the structure of chimera virus here (Zhang Y et al. Replication is the key barrier during the dual-host adaptation of mosquito-borne flaviviruses. *Proc Natl Acad Sci U S A.* 2022 Mar 22;119(12):e2110491119. doi: 10.1073/pnas.2110491119.).

However, dissemination of the CYV-ZIKV chimeric virus in *Aedes aegypti* and a high rate (75%) of transmission in orally inoculated mosquitoes is new and interesting and given that CYV has surprisingly been detected in a variety of mosquito genera previously provides insight on how it has reached multiple mosquito genera. The fact that WT CYV could be transmitted transovarially is also new, but not discussed. The fact that CYV is vertically transmitted and not CYV-ZIKV – are you suggesting that it is the structural proteins of the chimera that are blocking the vertical transmission of CYV-ZIKV?. Vertical transmission of a dISF has not been previously demonstrated.

Answer:

This is an excellent idea. Like other ISFs, such as *Aedes flavivirus* (AEFV) and *Parramatta River Virus* (PaRV), CYV could be transmitted transovarially (Haddow AD *et al.* First isolation of *Aedes flavivirus* in the Western Hemisphere and evidence of vertical transmission in the mosquito *Aedes (Stegomyia) albopictus* (Diptera: Culicidae). *Virology.* 2013 Jun 5;440(2):134-9; McLean BJ *et al.* The Insect-Specific Parramatta River Virus Is Vertically Transmitted by *Aedes vigilax* Mosquitoes and Suppresses Replication of Pathogenic Flaviviruses In Vitro. *Vector Borne Zoonotic Dis.* 2021 Mar;21(3):208-215). The fact that CYV is vertically transmitted but CYV-ZIKV is not suggests that structural proteins of the chimera play an important role in vertical transmission of CYV-ZIKV. Actually, we are working on this project, but it's out of the scope of this paper. We discussed this in the result part as “As an ISF, CYV could be efficiently transmitted via the transovarial route from

infected females to F1 mosquitoes. However, the transovarial transmission of CYV-ZIKV was almost abolished (Fig. 2h), implying that the structural proteins might be responsible for the vertical transmission of CYV. Therefore, the chance for CYV-ZIKV to spread to native mosquitoes is very low” in Page 4, line 146.

That the virus reaches sufficient titre in salivary glands and allows immunisation of mice through mosquito bites (3 mozzies x 3 times) is really surprising considering there could be little more (and probably less) than 1 ng of viral antigen present in the inoculum AND considering the virus does not show evidence of replicate in vertebrate cultures (and presumably the mouse). Nevertheless this was consistent with mice receiving a similarly low dose (10³ infectious units by IP with no adjuvant) also seroconverting with a neutralising antibody response. In this context further confirmation that there was no replication on the bitten/inoculated mice would have been useful here. Indeed, a further dissection of the mechanisms of the potency of the response to this virus in this context would have been far more compelling and useful than the claim that has provided a basis to control arbovirus transmission in sylvatic/zoonotic cycles (see below).

Answer:

Thank you so much for the comment. The CYV-chimera vaccine are highly immunogenic and consistent with previous papers using Binjari virus or Aripo virus vector to express the envelope proteins of ZIKV, dengue virus, or yellow fever virus (Hobson-Peters J, et al. A recombinant platform for flavivirus vaccines and diagnostics using chimeras of a new insect-specific virus. *Sci Transl Med.* 2019 Dec 11;11(522):eaax7888; Harrison JJ, et al. Chimeric Vaccines Based on Novel Insect-Specific Flaviviruses. *Vaccines (Basel).* 2021 Oct 22;9(11):1230.). We still can't explain why the tiny amount of CYV-ZIKV in the saliva can trigger strong immune response. But, there might be two explanations.

Frist, CYV-ZIKV and other dISF-vectored vaccines are between inactivated and live vaccines. In our previous paper, we found dISFs including CYV could enter vertebrate cells efficiently as ZIKV, but failed to initiate replication (Zhang Y et al. Replication is the key barrier during the dual-host adaptation of mosquito-borne flaviviruses. *Proc Natl Acad Sci U S A.* 2022 Mar 22;119(12):e2110491119. doi: 10.1073/pnas.2110491119.). Another dISF, Aripo virus, was demonstrated to be endocytosed into vertebrate cells and is highly immunomodulatory, producing a robust innate immune response despite its inability to replicate in vertebrate systems (Auguste AJ, et al. Isolation of a novel insect-specific flavivirus with immunomodulatory effects in vertebrate systems. *Virology.* 2021 Oct; 562:50-62. doi: 10.1016/j.virol.2021.07.004.). The strong innate immune response might enhance the humoral immune response.

Second, there are extensive researches suggest that vaccines delivered by microneedle patches can elicit better immune activation in the skin than one large needle (Manning JE, Cantaert T. Time to Micromanage the Pathogen-Host-Vector Interface: Considerations for Vaccine Development. *Vaccines (Basel).* 2019 Jan 21;7(1):10. doi: 10.3390/vaccines7010010.; Gurera D, Bhushan B, Kumar N. Lessons from mosquitoes' painless piercing. *J Mech Behav Biomed Mater.* 2018 Aug;84:178-187. doi: 10.1016/j.jmbbm.2018.05.025.). Thus, we think the mosquito biting injection behaves similar as the microneedle patches and can induce high immune response. But, we admit more research is needed to investigate the mechanism underlying.

According to the suggestion, we tested CYV-ZIKV replication in mice and no CYV-ZIKV replication was detected at bitten sites in the mouse skin (Fig. S9), which was consistent with inability to replicate at cellular level (Fig. 2k).

Supplementary Figure 9

Fig. S9. No replication of CYV-ZIKV in the local skin of mosquito bite site. The shaved skin in IFNAR^{-/-} C57/BL6 mice was bitten by 30 CYV-ZIKV-carrying mosquitoes per mouse for 20 min. The skin tissues at the bitten site were dissected after scarification 12, 24, 48, 72, or 96 h post-bite (n=3). The viral RNA of CYV-ZIKV in the skin was determined by real-time PCR.

Based on the immune response generated in the mosquito-bitten mice, it was no surprise that they became partially resistant to subsequent infection and failed to transmit the virus to feeding mosquitoes. This is interesting and novel data. However, the main message delivered by the authors appears to be that this is proof of concept for a “feasible” and “promising” use of this approach to immunise wildlife to interfere with sylvatic and zoonotic transmission cycles to prevent/reduce human infection, was far too speculative and optimistic in my mind and needs to be discussed with much more circumspection, if mentioned at all. There are just far too many variables for an approach like this work in a natural environment (multiple vectors of these viruses in sylvatic cycles with variable host preference, requirement for repeated release in remote regions, let alone getting regulatory approval for an uncontrolled release of GMO virus that is transmitted to animals (and humans

that get in the way). There was also no repeated passage of the virus in vertebrate systems to demonstrate the virus does not adapt to vertebrate replication through selected mutations.

Answer:

We agree that it needs a lot of efforts to evaluate the safety and efficacy of mosquito-delivered vaccines before field trial. In fact, there were several field trials using mosquitoes to control arbovirus transmission. Oxitec Ltd tested their GMO mosquito in Florida, US (Neuhaus CP. Community Engagement and Field Trials of Genetically Modified Insects and Animals. Hastings Cent Rep. 2018 Jan;48(1):25-36. doi: 10.1002/hast.808.). Dr. Zhiyong Xi’s team tested the *Wolbachia*-infected male *Aedes* mosquitoes in Guangzhou, China, from 2014 to 2017 and in Australia from 2017 to 2018 (Zheng X, et al. Incompatible and sterile insect techniques combined eliminate mosquitoes. Nature. 2019 Aug;572(7767):56-61. doi: 10.1038/s41586-019-1407-9.;

Beebe NW et al. Releasing incompatible males drives strong suppression across populations of wild and *Wolbachia*-carrying *Aedes aegypti* in Australia. Proc Natl Acad Sci U S A. 2021 Oct 12;118(41):e2106828118. doi: 10.1073/pnas.2106828118.). Thus, release of mosquitoes is rational for the control of arboviruses. Recently, a clinical trial was performed to test a genetically engineered *Plasmodium falciparum* parasite vaccine delivered by mosquito bites (Murphy SC, et al. A genetically engineered *Plasmodium falciparum* parasite vaccine provides protection from controlled human malaria infection. Sci Transl Med. 2022 Aug 24;14(659):eabn9709. doi: 10.1126/scitranslmed.abn9709.).

We added one paragraph in the discussion as “Release of sterile mosquitoes and *Wolbachia*-carrying mosquitoes have been studied for many years and under field trial with significant efficacy in controlling mosquito populations. Sustained releases of transgenic *A. aegypti* males with the OX513A lethal gene led to at least 80% suppression of the wild *A. aegypti* population in the Cayman Islands in 2010 and a suburb of Juazeiro, Bahia, Brazil in 2012. Releasing *Wolbachia*-infected male mosquitoes successfully reduced wild *A. albopictus* populations on two separate islands in Guangzhou, China, from 2014 to 2017 and wild *A. aegypti* populations in Australia from 2017 to 2018. Thus, mass production and release of mosquitoes is feasible for the control of arboviruses”.

We admit the mosquito-delivered vaccine is still at the early age of development. Many issues need to be addressed before releasing this kind of vaccine in a natural environment. We summarized these issues in the discussion as “The barriers limiting wildlife vaccination include: (i) involvement of multiple hosts in sylvatic transmission cycles; (ii) safety concerns for non-target species; (iii) high reproductive rates and population turnover; (iv) fastidious behaviors and difficulty in designing effective delivery systems; (v) difficult delivery due to extreme low or high population densities of the target hosts; (vi) environmental concerns for the release of genetically modified organisms; (vii) stability of a vaccine under prevailing environmental conditions; and (viii) low unit cost for vaccine purchase and delivery.”

Other than that, the mosquito-delivered vaccine could have more usages. The mosquito-delivered vaccine could also be applied to protect endangered animals such as Ruffed grouse decrease due to West Nile virus infection (Nemeth NM, et al. West Nile virus infection in Ruffed grouse (*Bonasa umbellus*) in Pennsylvania, USA: a multi-year comparison of statewide serosurveys and vector indices. J Wildl Dis. 2021 Jan 6;57(1):51-59. doi: 10.7589/JWD-D-19-00016. PMID: 33635996.). Recently, a clinical trial was performed to evaluate a genetically engineered *Plasmodium falciparum* parasite vaccine delivered by mosquito bites (Murphy SC, et al. A genetically engineered *Plasmodium falciparum* parasite vaccine provides protection from controlled human malaria infection. Sci Transl Med. 2022 Aug 24;14(659):eabn9709. doi: 10.1126/scitranslmed.abn9709. Epub 2022 Aug 24. PMID: 36001680.)

There was also no repeated passage of the virus in vertebrate systems to demonstrate the virus does not adapt to vertebrate replication through selected mutations.

Answer:

We have tested the stability of CYV-ZIKV in mosquitoes. As shown in Fig. 2k, CYV-ZIKV is very stable after 5 passages in mosquitoes. We also tried to passage CYV-ZIKV on vertebrate cells and

no infection was detected (Fig.2).

The infection barrier of dISFs in vertebrates has been studied extensively. Infection of two dISFs, BinJV and Aripo virus (ARPV), in vertebrates is mainly restricted in the post-entry step, which is likely mediated by the innate immune response or temperature (Imperato PJ. The Convergence of a Virus, Mosquitoes, and Human Travel in Globalizing the Zika Epidemic. *J Community Health*. 2016 Jun;41(3):674-9. doi: 10.1007/s10900-016-0177-7. PMID: 26969497; Ribeiro GS, et al. Influence of herd immunity in the cyclical nature of arboviruses. *Curr Opin Virol*. 2020 Feb;40:1-10. doi: 10.1016/j.coviro.2020.02.004. PMID: 32193135). Long Pine Key virus, another dISF, can't infect vertebrate cells due to entry and post-translational restrictions (Wikan N, et al. Zika virus: history of a newly emerging arbovirus. *Lancet Infect Dis*. 2016 Jul;16(7):e119-e126. doi: 10.1016/S1473-3099(16)30010-X. PMID: 27282424.).

In our previous study, we found Donggang virus (DONV) and CYV entered vertebrate cells as efficiently as the mosquito borne flaviviruses but failed to initiate replication. Their replication in vertebrate could be rescued by the exchange of the untranslated regions (UTRs) of those from Zika virus. And the barrier in virus assembly and secretion are still not identified. (Zhang Y, et al. Replication is the key barrier during the dual-host adaptation of mosquito-borne flaviviruses. *Proc Natl Acad Sci U S A*. 2022 Mar 22;119(12):e2110491119. doi: 10.1073/pnas.2110491119. PMID: 35294288).

Together, we speculate that it would be very difficult for dISFs to acquire the ability of replication in vertebrates.

Specific comments:

P3, para 3 – refs and full stop required at end of this sentence

“Chimeric ISFs expressing the envelope proteins of pathogenic flaviviruses do not replicate in vertebrates but can trigger a protective immune response in vertebrates”

Answer:

Thank you for pointing out. We added the references and full stop at end of this sentence in P3, para 3.

Several places - Binj change to Binjari;

Answer:

Thank you for pointing out. We changed the Binj to Binjari in line 332 and in line 348 page 8.

There needs to be much more detailed methodology, which can easily be provided as a supplementary file. In particular, greater detail is required for the following to allow accurate review of the data:

The neut method (below) is rather unusual and readout and interpretation criteria very sketchy for such a crucial part of the paper. There needs to be more detail and (published) evidence that it is comparable to standard neut assays in this context. Specifically, there also needs to be information on how many units of ZIKV were added into the neutralisation assays. The addition of NH₄Cl would

block the maturation of secreted virions, thus allowing for individual infected cells to be identified. I am not familiar with this type of assay.

The neutralizing activity of mouse sera was assessed using ZIKV MR766, Natal-RGN, and GZ01. Sera samples were three-fold serially diluted starting at 1:20 in DMEM with 2% FBS and 1% penicillin/streptomycin. The diluted sera were incubated with the same volume of ZIKV at 37°C for 30 min. The antibody and ZIKV mixtures were added to the Vero cells in 96-well plates for 2 h. Then, the mixtures were removed and replaced with DMEM plus 2% FBS, 1% penicillin/streptomycin, and 20 mM NH₄Cl. Vero cells were incubated at 37°C for 3 days. The neutralizing antibody was detected using the immunofluorescence assay described above.

Answer:

Thank you for pointing this out. This is a focus forming assay. The results were expressed as the FRNT50. We added more details as below.

The focus reduction neutralization test (FRNT) was performed to assess the neutralizing activity of mouse sera using ZIKV MR766, Natal-RGN, (GenBank sequence accession number, KU527068), and GZ01 (GenBank sequence accession number, KU820898) strains. The vero cells were seeded in 96-well plates at 8,000 cells per well at 24 h before the experiment. Sera samples were three-fold serially diluted starting at 1:20 in DMEM with 2% FBS and 1% penicillin/streptomycin. The diluted sera were incubated with the same volume of 100 FFU ZIKV at 37°C for 30 min. The antibody and ZIKV mixtures were added to the Vero cells in 96-well plates and incubated at 37°C for 2 h. Then, the mixtures were removed and replaced with DMEM plus 2% FBS, 1% penicillin/streptomycin, and 20 mM NH₄Cl. Vero cells were incubated at 37°C for 3 days. Then the cells were fixed, permeabilized and blocked as described in immunofluorescence assay. Virus foci were stained with anti-E antibody (4G2) followed by Alexa Fluor 488 labelled goat anti-mouse secondary antibody and counted under a fluorescence microscope. The results were quantified as the FRNT50.

In this assay we used mild basic NH₄Cl to block acidification of the endo-lysosomal network and the Golgi secretion network, thereby inhibiting secondary infection of flaviviruses, allowing individual infected cells to be identified. One fluorescent cell could be defined as one focus-forming unit. This method was modified from the fusion inhibitor assay applied in the alphavirus and flavivirus field (Liao M, Kielian M. Domain III from class II fusion proteins functions as a dominant-negative inhibitor of virus membrane fusion. *J Cell Biol.* 2005 Oct 10;171(1):111-20. doi: 10.1083/jcb.200507075. PMID: 16216925; Zheng A, Umashankar M, Kielian M. In vitro and in vivo studies identify important features of dengue virus pr-E protein interactions. *PLoS Pathog.* 2010 Oct 21;6(10):e1001157. doi: 10.1371/journal.ppat.1001157. PMID: 20975939).

This method is convenient and does not require complicated preparation of carboxymethylcellulose (CMC). CMC blocks the virus diffusion and NH₄Cl inhibits secondary infection. We tested the titer of our DENV stock side by side using these two methods and got similar results as below. The titer of DENV2 New Guinea C is 10^{6.8} FFU/ml using NH₄Cl and 10^{6.9} FFU/ml using CMC.

The TCID₅₀ immunofluorescence assay needs to be defined. Given that the readout is FFU, is this assay an immunoplaque assay? Has this method been published previously?

Answer:

The TCID₅₀ immunofluorescence assay is similar to the immunoplaque assay and has been used for determining virus titers previously (Nawtaisong P, et al. Effective suppression of Dengue fever virus in mosquito cell cultures using retroviral transduction of hammerhead ribozymes targeting the viral genome. *Virology*. 2009 Jun 4;6:73. doi: 10.1186/1743-422X-6-73. PMID: 19497123; Mishra P, et al. Antiviral Hammerhead Ribozymes Are Effective for Developing Transgenic Suppression of Chikungunya Virus in *Aedes aegypti* Mosquitoes. *Viruses*. 2016 Jun 10;8(6):163. doi: 10.3390/v8060163. PMID: 27294950). We added more details about the TCID₅₀ immunofluorescence assay. The readout is TCID₅₀/ml for the titer in saliva of 80 mosquitoes. The titer in saliva of single mosquito was detected by focus forming assay and the readout is FFU. We added the method of focus forming assay (page 9).

A Western blot is provided in Figure 1, with not methodology provided. Furthermore, there is no discussion on why the banding pattern of CYV-ZIKV differs from that of the WT ZIKV. I suspect that this could be differences in glycosylation, but it is not clear why WT differs from the chimera.

Answer:

Thank you for pointing this out. We provided the methodology of Western blot in page 12. The E protein of CYV has 499 aa and the molecular weight was predicted to be 54 kDa, while ZIKV E protein contains 504 aa and predicted to be 54.5 kDa. The E protein of ZIKV has one N-glycosylation site, while CYV has no N-glycosylation. We repeated the western blot. As show in Figure 1c, the E proteins of ZIKV and CYV-ZIKV displayed similar patterns with a main band at ~55kDa and some lower bands representing proteins with no glycosylation or some kind of degradation. CYV E protein showed one sharp band.

For the mouse immunization experiments, were the mice given chimeric virus that had been purified, or as a culture supernatant? Similarly for the mosquito experiments, were the mosquitoes provided purified virus, or virus as culture supernatant.

Answer:

For the mouse immunization experiments and for the mosquito experiments, the chimeric virus was from culture supernatant stored at -80°C, which had not been purified.

Please provide accession numbers for all of the ZIKV strains.

Answer:

We provide accession numbers for all the ZIKV strains. ZIKV MR766 (GenBank sequence accession number, HQ234498) in line 389 page 9, Natal-RGN, (GenBank sequence accession number, KU527068), and GZ01. (GenBank sequence accession number, KU820898) in line 511 page 11.

Method are missing for virus purification and growth kinetics.

Answer:

We provide the method of virus purification for western blot analysis for Figure 1c (page 12) and growth kinetics for Figure 1d (page 13) as below.

Western blot analysis of viral particles C6/36 cells were infected with CYV, ZIKV or CYV-ZIKV at an MOI of 1 and 10 ml of supernatants were collected at 3 dpi. Viral particles were precipitated by ultra-centrifugation through a 20% sucrose cushion by 39,000 rpm (SW41 rotor, Beckman, Fullerton, CA, USA) for 3 h at 4°C and pellets were resuspended in 100 µl PBS. Samples were separated by 10% SDS-PAGE and immunoblotted with anti-E monoclonal antibody 4G2 and the blots were scanned by Odyssey CLx.

Viral growth kinetics The C6/36 cells were passaged in a 6 cm dish at a density of 3×10^6 cells/dish. Twenty-four hours later, CYV or CYV-ZIKV was added to the cells at an MOI of 1. The supernatants were collected at 0, 24, 48, 96 hpi and stored at -80°C. The C6/36 cells were seeded in 96-well plates at 40,000 cells per well. Twenty-four hours later, virus supernatants were 10-fold serially diluted with RPMI medium containing 2% FBS and added to C6/36 cells (100 µl/well). After incubation at 28°C for 3 days, viral titers were determined by focus-forming assay.

Introduction – references are missing for other previously characterized Disf-based chimeric ZIKV vaccines – BinJV-ZIKV and ARPV-ZIKV

Answer:

Thank you for pointing this out. We have cited these papers in line 92 page 3. (Hobson-Peters J, et al. A recombinant platform for flavivirus vaccines and diagnostics using chimeras of a new insect-specific virus. *Sci Transl Med.* 2019 Dec 11;11(522):eaax7888. doi: 10.1126/scitranslmed.aax7888. PMID: 31826984; Harrison JJ, et al. Chimeric Vaccines Based on Novel Insect-Specific Flaviviruses. *Vaccines (Basel).* 2021 Oct 22;9(11):1230. doi: 10.3390/vaccines9111230. PMID: 34835160; Porier DL, et al. Enemy of My Enemy: A Novel Insect-Specific Flavivirus Offers a Promising Platform for a Zika Virus Vaccine. *Vaccines (Basel).* 2021 Oct 7;9(10):1142. doi: 10.3390/vaccines9101142. PMID: 34696250.)

Reviewer #3 (Remarks to the Author):

The manuscript from Wen, Ding et al reports on the use of mosquitoes infected with an insect flavivirus chimera as a novel vaccine delivery system. The results are noteworthy as they are the

first report of such a strategy and if the approach can be shown to be safe and effective could be used to reduce viral loads in animal reservoirs. The data presented is robust and convincing, but some additional work could improve the impact (discussed below). Methodology is sound and the authors should be congratulated on including the use of multiple ZIKV strains in the in vitro analysis and in vivo studies. Major and minor points listed below.

Q1 How dependent on the use of MR766 is the immunogenicity? CYV-ZIKV (MR766) had increased levels in the saliva suggesting altered tropism. This may be different for various chimeric viruses. Have the authors tested any other chimeric viruses? This would greatly improve the impact of the paper.

Answer:

This is a great suggestion. We created another chimeric virus expressing structure proteins of dengue virus 4. As shown in figure S6, the CYV-DENV4 can reach a titer peak of 10^8 FFU/ml in C6/36 as CYV-ZIKV. High titer was detected in the saliva after blood feeding. Thus, we speculate this would be a universal platform for many pathogenic flaviviruses.

Supplementary Figure 6

Fig. S6. Susceptibility of CYV-DENV4 in *A. aegypti*. (a) Titer of CYV-DENV4 in the C6/36 supernatant at 4 dpi (MOI=0.1). Con. indicates mock infected cells. (b,c) Females were blood-fed with CYV-DENV4 diluted to 1×10^8 FFU/ml. (b) The viral RNA level of the whole individual mosquito was detected by real-time PCR at 7 dpi. (c) The titer of CYV-DENV4 in the saliva of individual mosquito was detected by TCID₅₀ at 12 dpi.

Q2 The authors report a lower neutralization level for the mice exposed to the most mosquitoes (20 vs 10, Fig4J). Can the authors speculate on the reason for this? The memory response is also worse in the mice exposed to infected mosquitoes 3 times vs 2 times against the divergent ZIKV strains and should be mentioned and discussed.

Answer:

This is a great question. As shown in Fig. 4c, the immunogenicity of CYV-ZIKV was dose-dependent when administered by *i.p.* route. However, the immune response through mosquito bites was not dose-dependent. Saliva is a complicated mixture of proteins and small compounds which could suppress the immune responses to facilitate the infection of arbovirus. This might explain the lower neutralization level for the mice exposed to the 20 mosquitoes vs 10 mosquitoes and 3 times vs 2 times. In order to overcome this problem, we are managing to increase the CYV-ZIKV/saliva

ratio by enhancing the titer of CYV-ZIKV. We are also going to further study the effect of saliva on CYV-ZIKV immunogenicity. We discussed in Page 8, Line 339 as “We assume that components in the saliva might suppress the immune responses, which could be overcome by increasing the titer of CYV-ZIKV in the saliva”.

Q3 Are the particles effectively matured? The maturation state of expectorated virus should be assessed. This could be done by western blot, or alternatively the level of prM specific antibody could be evaluated in mice bitten by the CYV-ZIKV infected mosquitoes and compared to CYK-ZIKV immunized mice. Maturation state is important for ADE potential.

Answer:

This is a very good question. We expressed and purified the extracellular domain of ZIKV prM in bacteria. We selected sera sample with similar neutralizing activity from mice immunized by CYV-ZIKV-carrying mosquito in Fig 4j or CYV-ZIKV via *i.p.* route in Fig 4c. As shown in Fig S8, no obvious difference was detected as measured by ELISA using ZIKV prM coated plates. Thus, we speculate that the maturation state of CYV-ZIKV produced by C6/C36 cells or Aedes is similar. We added these results in Fig S8.

Supplementary Figure 8

Fig. S8. The level of ZIKV prM-specific antibodies in CYV-ZIKV-immunized mice by mosquito bites or via *i.p.* route. The purified ZIKV prM was detected by Coomassie Blue staining (a) and Western blot (b) by anti-His tag antibody. (c) The level of prM-specific antibody was detected by ELISA.

Q4 ADE should be evaluated. Do the mice become sensitive to dengue infection? Or does the sera from mice exposed to CYV-ZIKV cause ADE *in vitro*? This is very important when considering the targeted reservoirs will likely be exposed to more than one circulating flavivirus and multiple DENV serotypes.

Answer:

This is a very good point. We have evaluated ADE using an *in vitro* system as described in page 5. Briefly, The K562 cells were seeded in 24-well plates at a density of 10⁶ cells/well. Mouse sera were 3-fold serial diluted from 1:2 of in DMEM and incubated with equal volume of DENV (10⁵ FFU)

for 1 h at 37°C, then transferred to K562 cells. Viral RNAs in the supernatants were measured by real-time PCR at 4 dpi. As shown in Figure S7, CYV-ZIKV immunized mice sera showed no ADE activities against three dengue serotypes DENV1, 2 and 3.

Supplementary Figure 7

Fig. S7. No ADE of DENV mediated by CYV-ZIKV immunized sera in K562 cells. The K562 cells were seeded in 24-well plates (10^6 cells/well). Mouse sera were 3-fold serial diluted from 1:4 of in DMEM and incubated with equal volume of DENV 1 West Pacific strain, DENV2 New Guinea C strain or DENV3 H87 strain (10^5 FFU) for 1 h at 37°C, then transferred to K562 cells. Viral RNAs in the supernatants were measured using real-time PCR at 4 dpi.

ZIKV/Zika virus and CYV/Chaoyang are used interchangeably throughout – please use abbreviation once defined.

Answer:

We abbreviate Zika virus and Chaoyang virus after defined in this article.

Line 87 missing full stop and reference needed

Answer:

We corrected as suggested. We added the full stop at end of this sentence and references in line 92 page 3 as below. (Hobson-Peters J, et al. A recombinant platform for flavivirus vaccines and diagnostics using chimeras of a new insect-specific virus. *Sci Transl Med.* 2019 Dec 11;11(522):eaax7888. doi: 10.1126/scitranslmed.aax7888. PMID: 31826984; Harrison JJ, et al. Chimeric Vaccines Based on Novel Insect-Specific Flaviviruses. *Vaccines (Basel).* 2021 Oct 22;9(11):1230. doi: 10.3390/vaccines9111230. PMID: 34835160; Porier DL, et al. Enemy of My Enemy: A Novel Insect-Specific Flavivirus Offers a Promising Platform for a Zika Virus Vaccine. *Vaccines (Basel).* 2021 Oct 7;9(10):1142. doi: 10.3390/vaccines9101142. PMID: 34696250.).

Line 161 are/where quite low

Answer:

We deleted “are”.

Line 245 suggest reword “hardly”

Answer:

We change this sentence to “As expected, the mean ZIKV RNA levels in mosquitoes fed with the blood of CYV-ZIKV-immunized mice were lower than the detection limit, while most mosquitoes fed with the control mice (Con.) were positive for ZIKV” in Page 6, Line 260.

Line 280 define BinJ

Answer:

We defined BinJ as Binjari.

Line 306 “A lethal gene..” sentence is out of place and lacks context. At least include a reference.

Answer:

We deleted this sentence.

Line 324 second definition of mAb not required

Answer:

We deleted the second definition of mAb.

Line 329 The CYV virus genome was synthesized using...

Answer:

We corrected as suggested.

Line 437 the neutralization level was determined using immunofluorescence

Answer:

Thank you for pointing this out. We added more detail for Neutralization assay (page 11) as below. “The cells were then fixed, permeabilized, and blocked as described in immunofluorescence assay. Virus foci were stained with anti-E antibody (4G2) followed by Alexa Fluor 488 goat anti-mouse secondary antibody, and the fluorescence was observed under a fluorescence microscope. The results were quantified as the FRNT50”.

Fig S1 legend is confusing, assuming the authors mean the mosquitoes where infected with 1×10^8 /ml on line 710. Was this blood fed?

Answer:

We added more detail for Fig S1 legend as follows. Five- to six-day-old females were blood-fed with CYV-ZIKV diluted to 1×10^8 FFU/ml. Con. indicates control group that blood-fed with RPMI 1640.

Line 713, this is a mean across three independent experiments?

Answer:

This is not a mean across three independent experiments. We had done three independent experiments and the three independent experiments showed similar results. Here we showed the representative result of one experiment.

REVIEWERS' COMMENTS

Reviewer #2 (Remarks to the Author):

I thank the authors for the revisions made to their manuscript. I am satisfied that they have addressed my concerns.

I have only some additional minor corrections:

Line 28 - consider replication defective

Line 33 -leaking into the environment

Line 54 - originated

Line 100, for clarity, state that some or selected

ISFs, such as CYV can enter vertebrate cells.

Line 175 - there is a space missing between MR766 and by

Thank you for including the information in the methods that the chimera orally fed to the mosquitoes was as culture supernatant. Please also provide this information in the methods for the mouse immunization.

Reviewer #3 (Remarks to the Author):

The authors have adequately addressed questions raised in review. It would have been ideal to include a positive control for the ADE experiment. As this is not available it would be good to add a sentence in the discussion to highlight that ADE should be tested in vivo, particularly if the approach is broadened to DENV.

REVIEWERS' COMMENTS

Reviewer #2 (Remarks to the Author):

I thank the authors for the revisions made to their manuscript. I am satisfied that they have addressed my concerns.

I have only some additional minor corrections:

Line 28 - consider replication defective

Answer:

Thank you for pointing this out. We corrected as suggested in Line 24.

Line 33 -leaking into the environment

Answer:

We corrected as suggested in Line 29.

Line 54 - originated

Answer:

We corrected as suggested in Line 40.

Line 100, for clarity, state that some or selected ISFs, such as CYV can enter vertebrate cells.

Answer:

We corrected as suggested in Line 86.

Line 175 - there is a space missing between MR766 and by

Answer:

We added a space between MR766 and by in Line 161.

Thank you for including the information in the methods that the chimera orally fed to the mosquitoes was as culture supernatant. Please also provide this information in the methods for the mouse immunization.

Answer:

We provided this information as "from C6/36 supernatant" in the methods for the mouse immunization in Line 454.

Reviewer #3 (Remarks to the Author):

The authors have adequately addressed questions raised in review. It would have been ideal to include a positive control for the ADE experiment. As this is not available it would be good to add a sentence in the discussion to highlight that ADE should be tested in vivo, particularly if the approach is broadened to DENV.

Answer:

Thank you for pointing this out. We corrected as “with no ADE against DENV observed *in vitro*, which needs to be further confirmed *in vivo*.” in Line 295.